# Noninvasive electromyometrial imaging of human uterine maturation during term labor

Hui Wang [1,2,3,10], Zichao Wen [2,3,10], Wenjie Wu [2,3,4], Zhexian Sun [2,3,4], Zulfia Kisrieva-Ware[2,3], Yiqi Lin [2,3,5], Sicheng Wang [2,3,5], Hansong Gao [2,3,5], Haonan Xu[2,3], Peinan Zhao[3], Qing Wang[6], George A. Macones[7], Alan L. Schwartz[8], Phillip Cuculich[9], Alison G. Cahill[7] & Yong Wang [2,3,4,5,6] ✉

Electromyometrial imaging (EMMI) was recently developed to image the three-dimensional (3D) uterine electrical activation during contractions non-invasively and accurately in sheep. Herein we describe the development and application of a human EMMI system to image and evaluate 3D uterine electrical activation patterns at high spatial and temporal resolution during human term labor. We demonstrate the successful integration of the human EMMI system during subjects' clinical visits to generate noninvasively the uterine surface electrical potential maps, electrograms, and activation sequence through an inverse solution using up to 192 electrodes distributed around the abdomen surface. Quantitative indices, including the uterine activation curve, are developed and defined to characterize uterine surface contraction patterns. We thus show that the human EMMI system can provide detailed 3D images and quantification of uterine contractions as well as novel insights into the role of human uterine maturation during labor progression.

Two essential challenges in obstetrics worldwide are arrest of labor and preterm birth. Approximately 29% of women deliver via cesarean[1], the majority of which are performed for labor arrest[2]. Cesarean deliveries increase the risks of maternal morbidity and neonatal respiratory morbidity compared to vaginal delivery[3]. Preterm birth occurs in 10.6% of women globally[4], with increased infant risks of mortality before 5 years of age[5], adverse long-term neurodevelopmental outcomes[6], and an increased economic burden on the family and society[7]. Accurately assessing and interpreting uterine contractions is essential for diagnosing both labor dysfunction and preterm labor. In current clinical practice, overall signals generated by uterine contractions are measured qualitatively via tocodynamometry (TOCO) or quantitatively via an invasive intrauterine pressure catheter (IUPC)[8].

Previous studies have shown that these methods have limited ability to distinguish between women who will respond to induction/oxytocin augmentation and deliver vaginally and those that require cesarean[9]. Particularly, previous studies found that TOCO cannot identify patients who are in labor, at term or preterm[10,11]. In addition, between 30 and 50% of subjects diagnosed with preterm contractions go on to deliver at term[12]. Therefore, a better method capable of noninvasively imaging and quantifying uterine contractions is needed to address these clinical challenges.

To enable safe, noninvasive monitoring of uterine contractions, we recently developed a new imaging modality, electromyometrial imaging (EMMI), which noninvasively images the electrical properties of uterine contractions at high spatial and temporal resolution up to

[1]Department of Physics, Washington University, St. Louis, MO 63130, USA. [2]Center for Reproductive Health Sciences, Washington University School of Medicine, St. Louis, MO 63130, USA. [3]Department of Obstetrics and Gynecology, Washington University School of Medicine, St. Louis, MO 63110, USA. [4]Department of Biomedical Engineering, Washington University, St. Louis, MO 63130, USA. [5]Department of Electrical and Systems Engineering, Washington University, St. Louis, MO 63130, USA. [6]Mallinckrodt Institute of Radiology, Washington University School of Medicine, St. Louis, MO 63110, USA. [7]Department of Women's Health, Dell Medical School, The University of Texas at Austin, Austin, TX 78712, USA. [8]Department of Pediatrics, Washington University School of Medicine, St. Louis, MO 63110, USA. [9]Department of Cardiology, Washington University School of Medicine, St. Louis, MO 63110, USA. [10]These authors contributed equally: Hui Wang, Zichao Wen. ✉e-mail: wangyong@wustl.edu

2 kHz in sheep[13–15]. We demonstrated and validated that EMMI could accurately map electrical activity onto the entire three-dimensional (3D) uterine surface by comparing uterine electrical signals derived by EMMI from the body surface measurements (up to 192 electrodes) to those measured directly from the uterine surface in sheep[13]. Moreover, using the sheep model, experimental simulation studies mimicking noise contaminations anticipated in a clinical environment demonstrated that the electrical noise error within physiological ranges has a minor effect on EMMI accuracy[14,15].

Herein, we describe the development of the human EMMI system, for use in women in labor. This human EMMI system was employed to robustly image the 3D electrical activation patterns of uterine contractions from nulliparous and multiparous subjects in the active first stage of labor and demonstrates that EMMI can noninvasively characterize the initiation and dynamics of uterine electrical activation during uterine labor contractions. EMMI 3D maps and indices provide new insights into human myometrial electrical maturation, which is the development of the capacity of the uterus to appropriately initiate and conduct electrical signals across the myometrium. Thus EMMI's future clinical use will better characterize labor progression and facilitate labor management.

## Results

### Development and implementation of the human EMMI system

The human EMMI system incorporates subject-specific body-uterus geometry and multiple channel electrical measurements (up to 192 electrodes) from the body surface to reconstruct complete electrical activities in sequential frames across the 3D uterus.

In this study, human EMMI is performed on term labor subjects in three steps. First, a subject at ~37 weeks' gestation undergoes an MRI scan while wearing up to 24 MRI patches containing up to 192 MRI-compatible fiducial markers around the body surface (Fig. 1a). Second,

in the first stage of labor, when cervical dilation is at least 3 cm, and regular contractions are observed from the TOCO monitor, customized pin-type electrode patches are applied to the same locations on the body surface as the MRI fiducial markers. Because clinical devices such as a TOCO monitor and a fetal monitor must be applied to the abdomen to guide clinical decisions, the locations of the electrode patches were adjusted. To locate the electrode positions on an individual basis, an optical 3D scanner is used to record the actual electrode positions (Fig. 1b). Third, the subject undergoes body surface electrical recording (Fig. 1c). Each recording lasts ~15 minutes, and four recordings (up to one-hour total) are conducted for each subject in this study. The temporal sampling rate is 2048 Hz.

Raw data of MR images, optical 3D scanning, and body surface electromyograms (EMG) are preprocessed to generate a subject-specific body-uterus geometry (Fig. 1d) and body surface potential maps over the body surface (Fig. 1d). The body-uterus geometry includes the coordinates of the body surface electrode locations (blue dots in Fig. 1d) and the discretized uterine surface site locations (See body-uterus geometry construction in Methods). Filtering and artifact removal is applied to the raw EMG recording to improve the signal-to-noise ratio (See EMG signal preprocessing in Methods).

The Method of fundamental solutions[16] was employed to solve the three-dimensional Cauchy problem to generate uterine potential maps (electrical activity distribution on the uterine surface as a function of time every 10 milliseconds, Fig. 1f). These potential maps are essentially a 4D time-series data set: electrograms (Fig. 1g, D time series data over the entire recording period) at multiple sites on the uterine surface in 3D. During a contraction, we determine the uterine electrical activation times by measuring the start times of uterine electrogram burst (UEB) at each uterine surface site, which will be used to form the isochrone map (Fig. 1h). In the isochrone map, warm colors denote regions of the uterus that are electrically activated earlier during a

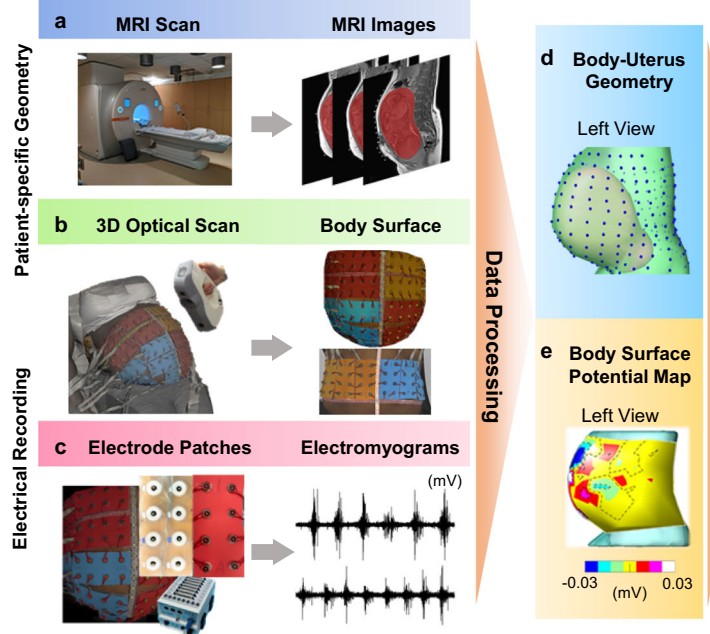
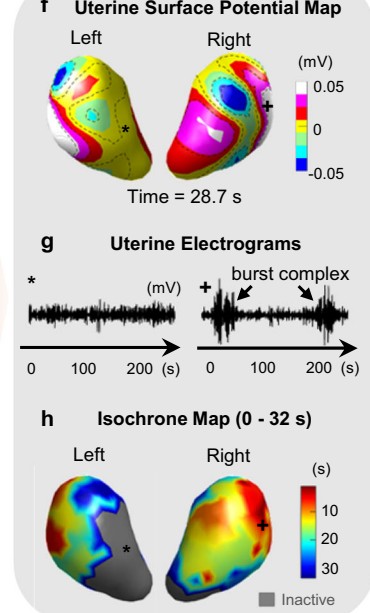

**Fig. 1 | The pipeline of the human EMMI system. a** An MRI scan is performed while the subject is wearing up to 24 MRI patches containing up to 192 markers. **b** An optical 3D scan of the body surface is performed while the subject is wearing electrode patches in the corresponding positions as the MRI patches. **c** Body surface electromyograms are simultaneously recorded from up to 192 pin-type unipolar electrodes assembled as patches. EMMI generates **d**, a body-uterus geometry from MR images with electrode locations on the body surface and, **e** a body surface potential map by rendering the electromyograms at each electrode at an

instant in time on the body surface. EMMI combines the two data sets to reconstruct **f** uterine surface potential maps (electrical activity across the uterus at a single time point). With the potential maps, we can generate the electrograms **g** electrical waveforms over time at each uterine site, and then derive **h**, the uterine region, and chronological sequence of electrical activation visualized as isochrone maps in 3D. 3D three-dimensional, MRI magnetic resonance imaging, EMMI electromyometrial imaging.

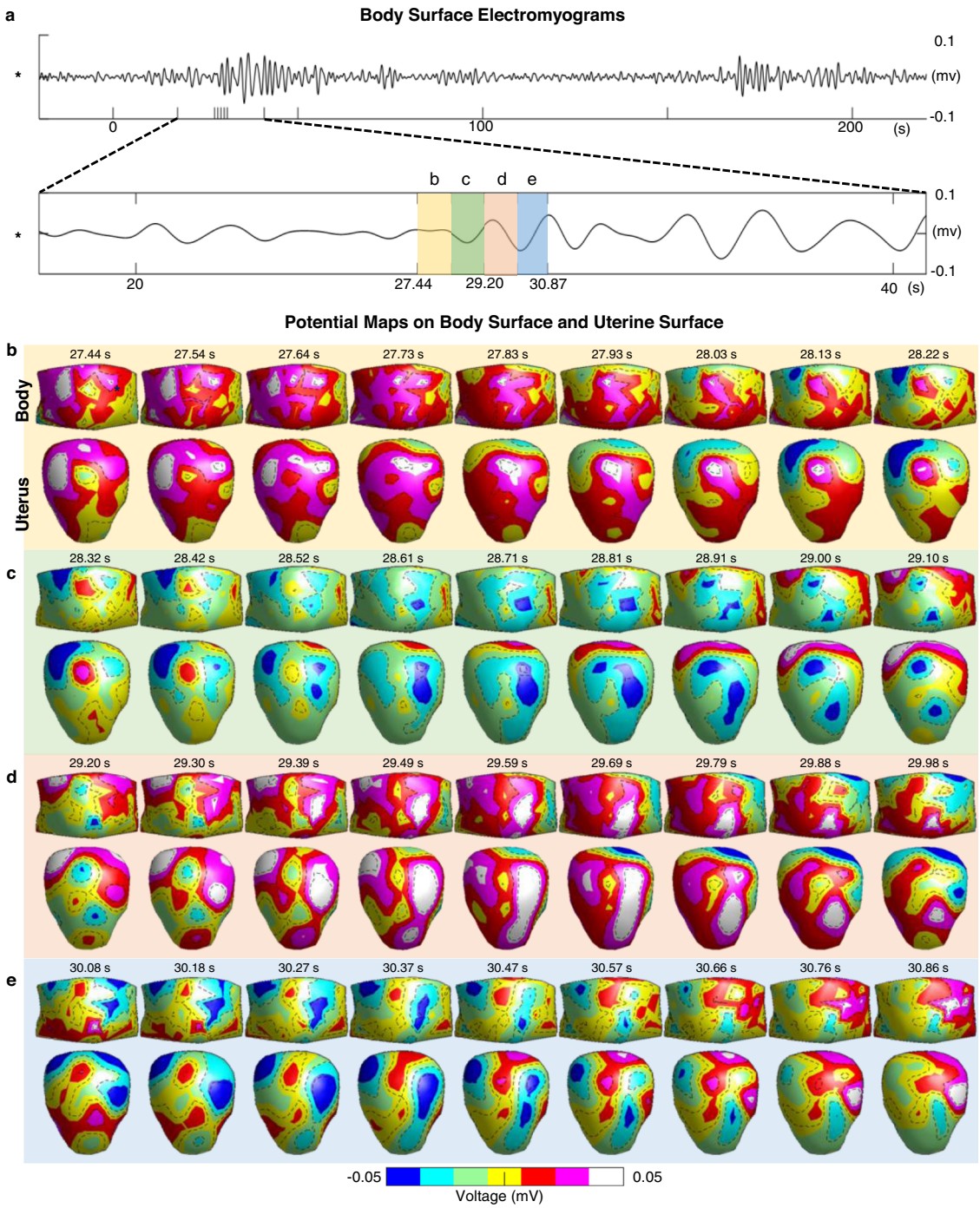

**Fig. 2 | Quantification of contractions with 3D uterine surface potential maps.**
**a** A representative body surface EMG was measured at the location labeled as star
(*) in Subject #2 experiencing a uterine contraction in the latent phase of labor. One
segment from -18th second to 41st second was magnified. **b**–**e** Sequential potential
maps on the body surface and uterine surface in anterior view at the indicated
times. Each row corresponds to the time window labeled as **b**–**e** in **a**, respectively.
EMG electromyogram. Source data are provided as a Source Data file.

contraction, cool colors denote regions that are activated later, and
gray regions that are inactivated. As shown in Fig. 1g and Fig. 1h, the
EMMI uterine surface electrograms reflect the local electrical activities.
For example, in Fig. 1g, the electrogram on the right has a UEB (where
UEBs can be detected above the baseline at an SNR > 5 Decibels) that
maps to a site (marked with a plus sign in Fig. 1h) in the isochrone map
near the region of the uterus that is first activated. In contrast, the
electrogram on the left does not show a UEB and maps to a site
(marked with an asterisk) in the isochrone map that showed no elec-
trical activity during the contraction. EMMI system (MRI scanner,

optical 3D scanner, and electrode patches) is described in detail in the
Methods section.

## Noninvasive imaging of human labor contractions
A body surface EMG measured from the body surface of one repre-
sentative subject (Subject #2) is shown in Fig. 2a. One recording seg-
ment from -18th to 41st second was magnified. Figure 2b through
Fig. 2e show sequential potential maps on the body surface and uterine
surface in the anterior view in the time windows labeled as b, c, d, and e
in Fig. 2a, respectively. At each indicated time window, the body

surface potential maps (body in Fig. 2b) were generated from the multichannel body surface electrode measurements. In comparison, the uterine surface potential maps (uterus in Fig. 2b) were reconstructed by EMMI. Unlike conventional EMG techniques in that the electrical activities are measured from the body surface, EMMI incorporates the subject-specific body-uterus geometry to generate sequential potential maps across the entire 3D uterine surface with high temporal resolution. The EMMI uterine surface potential maps enable noninvasive characterization of the electrical activities distributed over the entire uterine surface of a subject, and allow the detection of local electrical activities in the myometrium with high spatial resolution. Currently, the uterine surfaces consist of 320 vertices evenly, and the spatial resolution is ~2.5 cm.

The 4D spatial-temporal uterine surface potential map imaged by EMMI (in Figs. 2b through 2e) reveals the evolution of phase and magnitude inside a UEB. The uterine potential map can be reorganized into multichannel uterine surface electrograms based on the spatial locations on the 3D uterine surface. Each EMMI uterine surface electrogram reflects the local electrical activities of one uterine site. Based on the EMMI uterine electrograms during a uterine contraction, we define the electrical activation on the uterine surface by the initiation of a UEB. The term "uterine electrical activation time" or just "activation time" is used here to refer to the initiation time of the UEB. For the same representative subject shown in Fig. 2, simultaneous TOCO signal and five representative uterine surface electrograms (A–E in Fig. 3b) from five uterine surface sites (Fig. 3d) during two consecutive uterine contractions were shown in Fig. 3a, b, respectively. For each EMMI uterine surface electrogram, UEBs were detected (Supplementary Fig. 2), labeled by the upward red step lines, and overlaid on the electrograms. The rising edge of the red step line indicates the activation time during the contraction (green arrows). During the first uterine contraction, uterine surface electrograms from sites A through D demonstrated clear UEBs, suggesting that uterine sites A to D were electrically activated. In comparison, no UEB was detected in the uterine electrogram from site E, indicating that the myometrium around site e was inactive. Thus the entire myometrium was not electrically active and contributed to the uterine contractions. Inspecting all uterine surface sites, the earliest activation and the latest activation times can be detected and are marked by the dashed black vertical lines in Fig. 3a, b. EMMI uterine surface electrograms also suggest that the activation sequence among different uterine sites can change from one contraction to the next. For example, site B activated earlier than sites C and D in the first contraction, while site B activated after sites C and D in the second one (Fig. 3a).

The detailed sequential activation process during the first contraction was demonstrated in Fig. 3c. The upper row shows the sequential uterine isochrone maps at different times. The warm-colored (red and yellow) regions activated early, the cool-colored regions (cyan and blue) activated late, and the gray-colored regions were not activated. The lower row shows the activation ratio (AR), defined as the percentage of uterine regions that were activated at times associated with each uterine map above. AR is calculated as dividing the area of the activated uterine region by the total uterine area as a function of time. At the end of the activation process, the complete isochrone map of uterine activation (Fig. 3d) was generated to visualize the electrical activation pattern during the entire uterine contraction. The isochrone map reveals a complete 3D activation sequence, which does not show clear long-distance propagation. First, EMMI can detect the active or inactive region during a contraction. When a large portion of the uterus remains inactive, there is insufficient myometrium to support long-distance propagation. Second, even when a large portion of the uterus is active, we did not find cardiac-like long-distance propagation within the activated region.

Based on the rich spatial and temporal information in the isochrone map, an EMMI activation curve can be generated to reflect the temporal change of the AR over time during the entire uterine contraction period (Fig. 3e). The morphology of the EMMI activation curve reflects multiple key features of uterine contraction. Maximal activation ratio (MAR) can be quantified as the total activated myometrium by the end of the contraction. The activation curve slope (ACS) is defined to reflect the slope of the activation process (black dashed line in Fig. 3e), defined as MAR divided by the time taken to reach MAR during a contraction. Based on the activation curve, the initial 33% of the active myometrium regions can be detected and defined as early activation regions and mapped back onto the 3D uterine surface to form the early activation map (Fig. 3f). In the early activation map, early active regions were shown in red, late active regions were shown in blue, and inactive regions were shown in gray (Fig. 3f). The fundus area (the 25% of the uterine surface area in the anatomical superior uterine segment, see Method) was labeled by the white dashed line (Fig. 3f), and the fundal early activation ratio (FAR) is defined as the percentage of early activation region located within the fundus area. FAR measures the extent of fundal myometrium involved in the early activation during a contraction.

## Imaging uterine contractions during labor in nulliparous patients

EMMI was employed to study five nulliparous subjects (Subjects #1–5) in the active phase of term labor (Fig. 4). Subject #1 was imaged by EMMI when her cervical dilation was 3.5-4 cm (Fig. 4a). The prominent activation feature of the isochrone maps is that the activated myometrial regions were small (the gray indicates the inactive myometrium. MAR: 6.25%, 8.13%, and 19.38%) and primarily distributed at the middle and lower segments of the uterus. Based on the isochrone maps, the uterine activation curves were derived (blue curves in Fig. 4a; see details in Fig. 2e). For Subject #1, the uterine contraction activation curves were flat. The subject's ACS values were low (0.25%/s, 0.22%/s, and 0.34%/s), and FAR values were zeros. The EMMI isochrone maps and indices suggested that the subject's uterus was not yet electrically mature nor strongly engaged in generating forceful, synchronized contractions during the period of the electrical recording. It took 7.01 h for the subject to reach full cervix dilation after the electrical recording, and the average cervical dilation rates were 0.86 cm per h. Combined with the subject's clinical data, EMMI findings suggest that uterine contractions involve a small amount of myometrium, as indicated by the low MAR, at the early stage of active labor.

For Subject #2 (Fig. 4b), three representative contractions were imaged at 4.07 h before full cervix dilation. The subject's cervical dilation remained at 4 cm unchanged throughout the electrical recording. Although the entire uterus was not activated during the contractions, as shown in the isochrone maps, the activated uterine regions are much greater than in Subject #1. The MAR has a higher value than those in the first subject, suggesting a higher value of MAR (38.75%, 48.44%, 50.31%). Interestingly, the MAR is increasing across different contractions during the electrical recording period, suggesting that the uterus is actively recruiting more myometrium (e.g., 12.56% more myometrium was recruited into the third contraction than in the first). Within the activated uterine regions, different activation sequences were imaged for the three contractions, and no fixed early activation regions were observed. However, all three contractions were activated from the anterior-inferior fundus and lateral areas of the uterus (red-yellow regions). Activation curve slope (ACS) also increased from 1.19%/s to 1.37%/s, representing a 0.18%/s increase, suggesting the active uterine regions contracted in a more synchronized fashion in the later contractions. Fundal early activation ratios (FAR) were 7.5%, 1.25%, and 36.25%, respectively, suggesting a more fundus-initiated uterine contraction in the third contraction.

The cervical dilation of Subject #3 (Fig. 4c) was maintained at 5 cm during the electrical recording, which is 1 cm larger than that in Subject #2 (Fig. 4b). Three representative contractions were imaged at 14.41 h

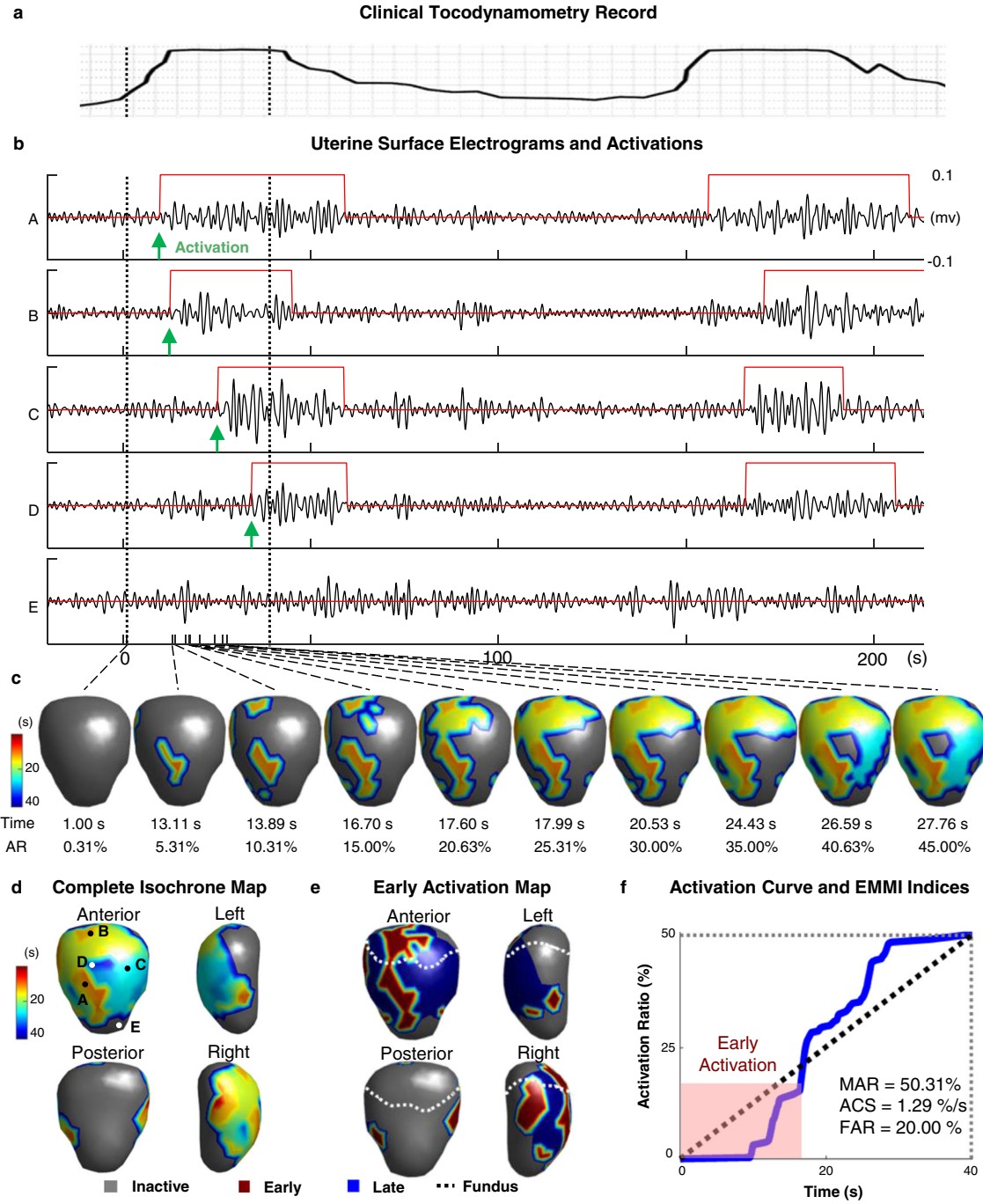

**Fig. 3 | Quantification of contractions with electrograms on the uterine surfaces. a** For a contraction in Subject #2 that the clinical TOCO monitor confirmed, **b** the electrical activations are defined as the initiation of UEB in the multichannel electrograms at about 320 uterine sites covering the entire uterine surface. In the 5 representative uterine electrograms from the indicated sites marked A through E in **d**, red step lines denote the UEB, green arrows denote the electrical activations, and the dashed black lines denote the earliest and latest electrical activations. **c** The entire activation process is visualized by the generation process of the uterine activation isochrone map and the activation curve. The former shows the activation location and time across the 3D uterine surface; the latter shows how the activation ratio increases over time. It occurs during the first part of the TOCO signal of contraction. **d** The complete isochrone map reflects the electrical activity of the

myometrium in time and space during the contraction, where warm colors denote uterine regions that are activated early, cool colors denote the regions that are activated late, and gray denotes the regions that are never inactivated. **e** In the early activation map, inactive regions are in gray, and activated regions are divided into two parts: early activation (red) the 33% of areas that are activated first in time, and late activation (blue) the remaining 67%. The fundal boundary is labeled as a dashed white curve. **f** The activation curve reflects the temporal progression of the electrical activation during the contraction. EMMI indices (MAR, ACS, and FAR) quantify the electrical properties of the myometrium. TOCO tocodynamometry, UEB uterine electrogram burst, AR activation ratio, EMMI electromyometrial imaging, MAR maximal activation ratio, ACS activation curve slope, FAR fundal early activation ratio. Source data are provided as a Source Data file.

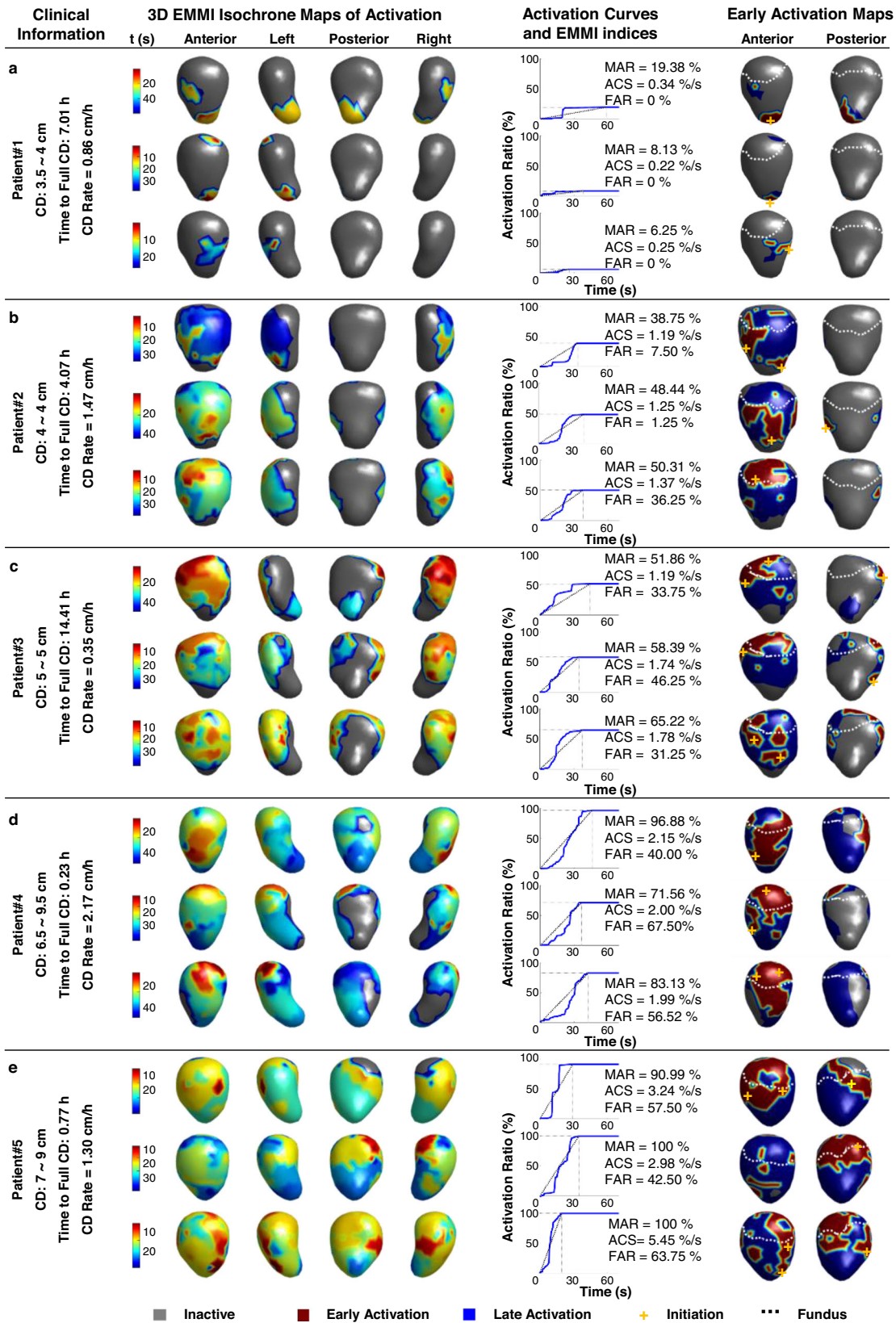

before full cervix dilation. However, the time to full dilation was 14.41 h, and the cervical dilation rate was 0.17 cm per h, indicating that this subject was in the latent phase of labor and had slow clinical labor progress. Similarly, we did not observe identical activation patterns and the existence of consistent initiation sites in Subject #2. Specifically, Subject #3 has high and increasing MAR for the three

contractions (51.86%, 58.39%, and 65.22%). Similarly, the ACS also increased dramatically in the last two contractions (1.19%/s, 1.74%/s, and 1.75%/s). It was also noted that the early activation region (red-yellow) was dominantly located in the fundus region for the three contractions (FAR: 33.75%, 46.25%, and 31.25%), suggesting fundus-initiated contractions in this subject during the electrical recording.

**Fig. 4 | EMMI activation patterns of uterine contractions during active labor in nulliparous women. a** In Subject #1, the cervical dilation changed from 3.5 cm to 4 cm during the electrical recording and the cervix fully dilated to 10 cm 7.01 h after the recording was completed. The cervical dilation rate was calculated at 0.86 cm per h. 3D EMMI isochrone maps for three representative contractions were shown in four views. The uterine regions in warm colors were activated earlier, cool colors regions activated later, and gray regions were inactivated. The color bar on the left denotes the activation time. The activation curve and associated EMMI indices were derived from each contraction. The early activation map highlighted the early activations (earliest 33% percent of activation, red), and the fundal area was labeled by a white dashed line. **b**–**e** Results for Subjects #2, #3, #4, and #5. Same format with Subject #1. 3D three-dimensional, EMMI electromyometrial imaging, CD cervical dilation, MAR maximal activation ratio, ACS activation curve slope, FAR fundal early activation ratio. Source data are provided as a Source Data file.

Despite the strong uterine contractions, the subject's cervix dilated at a very slow rate of 0.17 cm per h to the full cervical dilation after the electrical recording.

EMMI was used to examine two subjects with cervical dilation greater than 5 cm (Subjects #4 and #5). Subject #4 (Fig. 4d) was mapped at 6.5-9.5 cm cervical dilation and 0.23 h before full dilation, dilated at the rate of 2.17 cm per h, which was much faster than that observed in Patient #1–3. The uterus was highly active (MAR: 96.88%, 71.56%, and 83.13%), fairly synchronized (ACS: 2.15%/s, 2.00%/s, and 1.99%/s). and had high FAR (40%, 67.50%, and 56.52%). The MAR values of those contractions are much higher. Similar observations were made for Subject #5 (Fig. 4e). Considering that both Subjects #4 and #5 have strong uterine contractions suggested by high MAR, ACS, and FAR, the significant difference in cervical dilation rates between the two subjects suggested an inter-subjects difference in cervix properties as we observed for subjects earlier in active labor (Fig. 4b, c).

### Imaging uterine contractions during labor in multiparous patients

Five multiparous subjects were imaged by EMMI (Fig. 5). Similar to the findings in the nulliparous subjects, no fixed initiation sites or consistent activation patterns during uterine contractions were observed. In contrast to the uterine contractions imaged at the early stage of active labor in nulliparous subjects (Fig. 4a, b), EMMI found larger MAR, ACS, and FAR values in the uterine contractions from multiparous subjects (Fig. 5a, b). Subject #6 was mapped at 4 cm cervical dilation, 6.95 h before full dilation, and progressed at 0.86 cm per h after mapping. The MAR (38.75%, 52.50%, and 73.44%) were high and increasing. The ACS (0.91%/s, 1.20%/s, and 2.11%/s) and FAR (6.25%, 12.50%, and 58.75%) followed the same trend as the MAR, indicating the subject was experiencing uterine contractions with quickly increasing strength during the period of the electrical recording. In Subject #7, the cervical dilation range was 4-4.5 cm, and the time to full dilation was 1.62 h. Her labor progressed at dilation rates of 3.39 cm per h. The MAR (50.93, 40.68%, and 42.86%), ACS (2.37%/s, 1.41%/s, and 1.02%/s), and FAR (36.25%, 3.75%, and 15.00%) were high in this subject. In the early stage of active labor, EMMI found that the uterine contractions in the multiparous subjects seem stronger than those in the nulliparous subjects, which may suggest earlier electrical maturation.

At the later stage of active labor in the multiparous subjects (Fig. 5c–e), the MAR, ACS, and FAR values were not significantly increased compared to the nulliparous subjects (Fig. 4c–e). Subject #8 was mapped at 5 cm cervical dilation, 3.47 h before full dilation, and progressed at 1.44 cm per h after mapping. The MAR (20.94%, 25.63%, and 25.31%) was <30%. The ACS (0.43%/s, 0.43%/s, and 0.49%/s) were small and FAR (16.25%, 21.25%, and 8.75%) were normal. In Subjects #9 and #10, labor progressed at dilation rates of 2.47 cm per h and 3.52 cm per h, which are ~2.5 and 3.4 times faster than the average rate of 1 cm per h for 90% of the population. The cervical dilation ranges were 5-6.5 cm and 6.5-8 cm, and the time to full dilation was 1.42 h and 0.57 h, respectively. The MAR (15.63%, 19.38%, and 35.63%; 10.31%, 26.25%, and 31.25%) was <40% in both cases. The ACS (0.40%/s, 0.66%/s, and 1.09%/s; 0.26%/s, 0.54%/s, and 0.46%/s) was small and the FAR (6.25%, 7.50%, and 35%; 10.00%, 21.25%, and 31.25%) were normal. These findings may suggest that uterine contractions with lower MAR at the later stage of active labor are sufficient to remodel the cervix effectively and rapidly in the multiparous subjects.

## Discussion

In addition to the TOCO monitor and intrauterine pressure catheter (IUPC), multiple research tools have been developed and evaluated to study and evaluate uterine contractions. Magnetomyography (MMG) detects the subtle uterine magnetic activities with an array of superconducting quantum interference devices (SQUID)[17], including 151 sensors arranged in a fixed pattern to collect signals from the anterior abdominal region without much attenuation and distortion from the interfaces in volume conductor[18,19]. Although MMG data correlate with contractile events perceived by mothers and provide distribution maps of local uterine activity, this method does not provide a three-dimensional view of the entire uterus. It requires a large piece of specialized equipment in a magnetically shielded room to measure the weak MMG signals. Electromyography (EMG, also called electrohysterography, EHG) has been developed as an alternative way to noninvasively monitor the uterine electrical activity underlying contractions via several electrodes placed on the anterior part of the abdomen[20,21]. These EMG signals measured on the body surface are the spatial integral of action potentials from the underlying uterine smooth muscle (myometrial) cells[22]. EMG can be used to generate a uterine activity tracing that emulates TOCO and is more reliable than TOCO in patients with obesity[23], providing objective measurement of regional electrical activities. Specifically, EMG studies have revealed that the electrical propagation velocity increases at active labor compared to that at non-active labor[24–26]. Although some EMG studies have revealed that the electrical propagation velocity increases at active labor compared to that at non-active labor[27], the previous studies also reported controversial findings. Additionally, several features of EMG signals, such as the intensity, peak frequency of power spectrum, etc., show some promise in identifying the onset of labor[28–36]. Although the development of EMG has shed light on electrical activation during contractions, EMG is still limited to measuring a small area on the maternal abdomen and does not have sufficient spatial coverage and specificity to reflect the electrical activation pattern on the entire 3D uterine surface. This may well explain the complex and heterogenous electrical activity propagation patterns measured with limited number and non-standard configuration of the body surface electrodes in subjects during active labor[27,37–39]. In theory, one can overcome the limitations of body surface EMG by placing electrodes directly on the uterine surface. However, those invasive studies are usually challenging to conduct in animals and unethical in humans.

To address these limitations and inspired by the success of the sheep EMMI system[13–15], we developed a human EMMI system and demonstrate its unique advantages, i.e. noninvasively image the electrical activities underlying uterine contractions during labor across the entire 3D uterine surface in high spatial and temporal resolution (Fig. 1). Uterine surface potential maps imaged by EMMI provide a noninvasive measure of uterine contraction patterns without the need for surgery and placement of uterine surface electrodes (Fig. 2). EMMI can also image the uterine surface electrograms to reflect the local uterine electrical activities (Fig. 3). Based on the morphology of the EMMI uterine surface electrogram across the entire uterine surface with high spatial resolution, the inactivated myometrium regions can

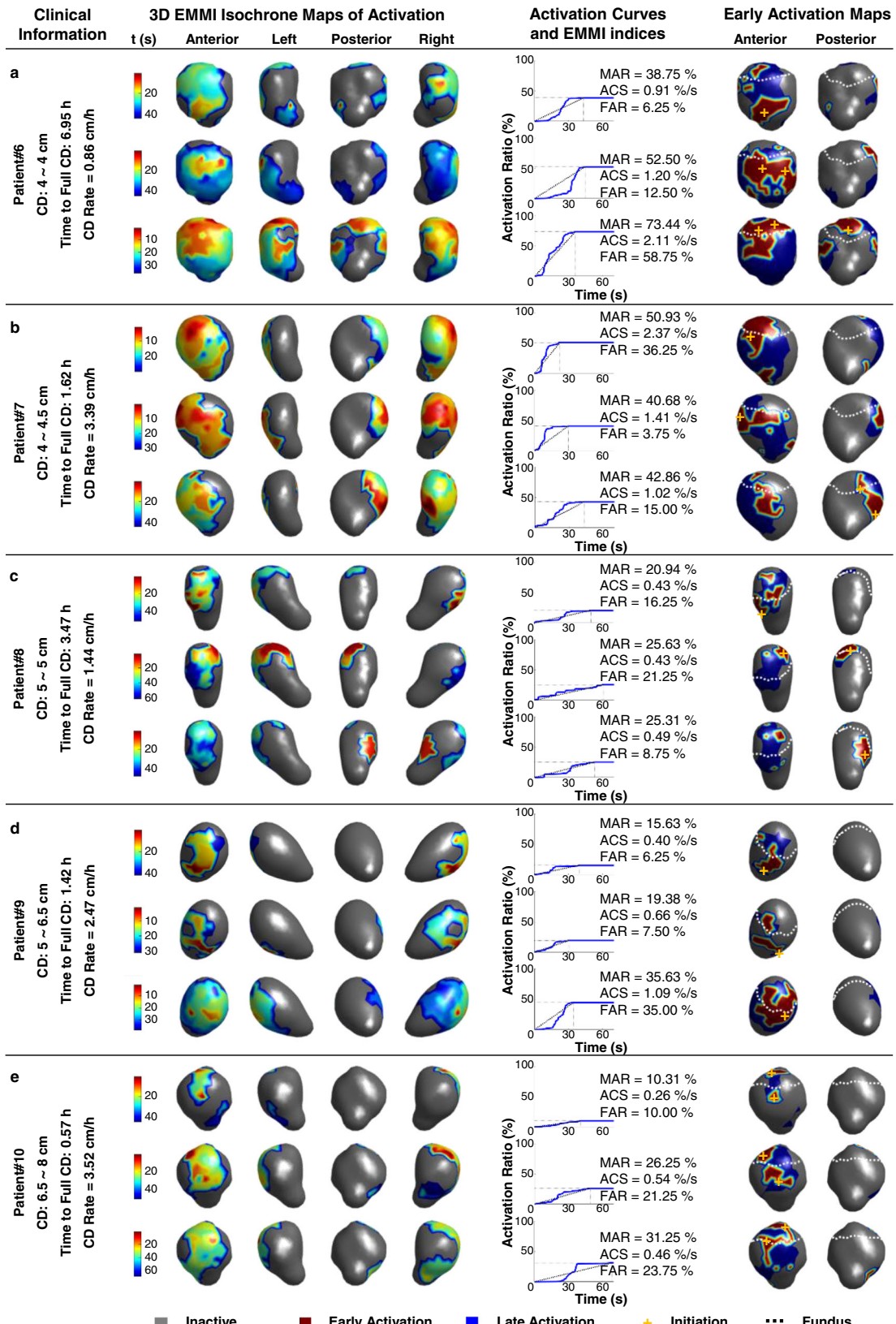

**Fig. 5 | EMMI activation patterns of uterine contractions during active labor in multiparous women. a** In Subject #6, the cervical dilation remained at 4 cm and reached 10 cm in 6.95 h after the EMMI recording was completed. **b–e** Results for Subjects #7, #8, #9, and #10. Same format with Subject #6. 3D three-dimensional, EMMI electromyometrial imaging, CD cervical dilation, MAR maximal activation ratio, ACS activation curve slope, FAR fundal early activation ratio. Source data are provided as a Source Data file.

be well delineated, which cannot be quantified by conventional body surface EMG (Fig. 3). For the activated uterine regions, the derived activation sequence (isochrone map) reflects the uterine contraction pattern (Fig. 3). Based on our EMMI findings in both nulliparous and multiparous subjects, the entire uterus is not active during uterine contractions, especially in the early phase of the labor. This cannot be detected by the conventional TOCO or body surface EMG (Figs. 4 and 5).

Distinct from rhythmic cardiac activation, uterine contraction has dramatically different activation patterns from contraction to contraction. This is consistent with previous findings of the absence of a predominant propagation direction[40,41]. Our results indicate that there are no consistent early activation regions in different uterine contractions, and this is direct evidence against an anatomically fixed, cardiac-like pacemaker region in the human myometrium[41]. Finally, long-distance propagation of activation is also not evident in human uterine contraction[41–43]. This human EMMI study thus demonstrates that EMMI can noninvasively provide rich information on human uterine activation patterns.

EMMI isochrone maps provide detailed activation sequences over the entire activated uterine regions. Based on the information in the 3D isochrone map, the activation curve describes the percentage of activated uterine area at different over the course of a contraction (Fig. 3) and reflects the temporal progression of myometrium activation. Two EMMI indices, maximal activation ratio (MAR) and activation curve slope (ACS), can be defined (Fig. 3) from the uterine activation curve. Particularly, MAR indicates the total surface area of the uterus that becomes electrically active during an individual contraction; and ACS indicates the rate of the uterine electrical activation development. Although we observed the overall monotonic increase of MAR and ACS during the electrical recording period of the active phase of human labor, MAR and ACS may temporally decrease and fluctuate in some cases (Fig. 4e, Fig. 5b). This can be potentially explained by the presence of the refractory regions generated by the complicated and dynamically changing activation sequences during uterine contractions. In addition, based on the anatomy of the uterine fundus and the early activations (the 33% active uterine sites that activate first), we define the third EMMI index, fundal early activation ratio (FAR) (Fig. 3). Although no fixed, cardiac-like pacemakers were observed by EMMI in our study, FAR objectively quantified the percentage of the fundal region that contributed to the early activation and generated contractions to dilate the cervix. In comparison to the overall increasing trend of MAR and ACS during labor, FAR demonstrated more contraction-to-contraction variations. This is consistent with the dynamic nature of uterine activation patterns and can be potentially used as a novel imaging biomarker to reflect the uterine contraction pattern during normal and arrested labor[2,44]. As described herein, we have developed a human EMMI system and imaged human uterine contractions during active labor in 10 subjects (5 nulliparous and 5 multiparous subjects). With the uterine activation pattern images and newly defined indices, EMMI will allow us to "see" and quantify the uterine contractions in 3D for each individual subject during labor and provide new insights about human labor progression beyond cervical dilation. In the nulliparous subject group, EMMI observed weaker, less synchronized, and less fundus-dominated contractions at the early stage of active labor. The uterine contractions became much stronger and synchronized in the later stage of active labor. It is known that both cervical properties (such as stiffness, etc.) and uterine contractions affect labor progression. Poor labor progression or labor arrest can be caused by insufficient uterine contraction, stiff cervix, or both. The EMMI isochrone maps, early activation maps, activation curves, and indices provide new quantitative characteristics of the uterine contributions to labor progression. This will be critical to clinically confirm or exclude the uterine causes of slow labor progression. For example, the slow labor progression in Subject #3 was not caused by

the inactive uterus, but potentially by a stiff and slowly-dilating cervix (Fig. 4c). In the multiparous subject group, stronger uterine contractions were indicated by the larger MAR, ACS, and FAR values were observed in the early stage of the active labor. This suggests the faster electrical maturation of the uterine contraction compared with that of nulliparous subjects. This finding supports the "myometrium memory" effect in multiparous subjects[43,45,46]. In addition, EMMI found uterine contractions with lower MAR at the later stage of active labor in the multiparous subjects in comparison with the nulliparous groups. However, the weaker contractions seem sufficient to dilate the cervix at a faster rate than that observed in the nulliparous group. This finding suggests that the cervix of multiparous subjects may be generally softer and easier to be dilated than the cervix of nulliparous subjects, consistent with what is clinically observed. The clinical utility of EMMI-derived outputs will require prospective clinical trials involving multiple EMMI assessments throughout labor, for example, the correlation of MAR with cervical dilatation. In this initial study with the 10 uncomplicated singleton pregnancies studied with EMMI in labor at 37w0d – 40w6d gestation and at cervical dilatation between 3.5 and 9.5 cm, we observed that nulliparous subjects demonstrated different phenotypes from the multiparous subjects (Figs. 4 and 5). These initial results may be suggestive of subgroups of uncomplicated nulliparous women in labor.

In this first human EMMI study, we derive the intuitive EMMI indices, such as MAR, ACS, and FAR, which relate to organ-level electrical activity patterns and propagation. We found that the EMMI-derived human uterine activation curves have a sigmoidal evolution nature with respect to time, which potentially reflects the bioelectrical stimulation-response dynamics during uterine contractions[47]. An earlier in vitro study using the rabbit uterine muscle studied the relationship between maximum contraction tension and duration/strength of electrical stimulation[48]. They found Hill's force-velocity relationship between the kinetics of isotonic contractions as a function of load (tension). EMMI-derived human uterine activation curves clearly confirm a potential regenerative process in human uterine contractility using a positive feedback mechanism. These findings will enable future studies to derive a comprehensive and uterus-relevant physiological assessment of human uterine contractions using EMMI.

Although we focus on the technological development and scientific benefits of the EMMI system in this work, EMMI holds great potential for wide clinical application. Pitocin, a synthetic version of naturally derived oxytocin, is the most commonly used medication to augment labor progress. However, oxytocin is released in a pulsatile fashion, while Pitocin is given in an escalating, continuous drip, titrated to contraction frequency and intensity. But nothing is known regarding the direct effect of Pitocin on myometrial electrical activation and propagation, and whether or not this non-physiologic delivery is optimal has never been explored. EMMI will enable these explorations and uncover further insights regarding the physiology of labor and labor management.

There are several limitations to the current human EMMI system and analyses that can be addressed and improved in future work. Currently, a large number of BioSemi Active electrodes (BioSemi B.V, Amsterdam, The Netherlands) are placed around the subject's body surface. In order to speed up electrode application, increase the comfort of wearing those electrodes, and decrease the cost, we are currently developing low-cost, elastic, disposable electrode patches using printing technology[49–51]. Optimization of electrode number and distribution will also increase subject compliance with the EMMI system. Another factor limiting the scalability and accessibility of the current human EMMI system is the availability and cost of MRI. A portable and low-cost ultrasound imaging system that is widely available in obstetric clinics can be integrated with the EMMI system to acquire the subject-specific body-uterus geometry. The current EMMI study only imaged uterine contractions for 1-hour from a small number

of subjects. Longer electrical recordings covering the entire labor process from a larger normal-term labor subject cohort will enable us to build the "normal term atlas" describing the detailed electro-physiology and normal standards of uterine contractions at high temporal and spatial resolution. This atlas will facilitate translational studies to define the mechanisms underlying normal human labor and identify EMMI uterine contraction indices and the spatial-temporal signatures of uterine contractions that may be altered in subjects with labor arrest and preterm contractions. In the longer term, this atlas could be used in clinical trials aimed at testing interventions to prevent labor complications such as labor arrest, preterm birth, and post-partum hemorrhage.

## Methods

### Study implementation

This study was approved by the Washington University Institutional Review Board (No. 201612140). Nulliparous and multiparous subjects were recruited, and those willing to participate underwent informed consent. The subjects were compensated 50 USD for MRI and 50 USD for labor mapping in the form of prepaid gift cards. In this manuscript, deidentified data from 10 term subjects are presented, among which 5 are nulliparous and 5 multiparous. The ages of the subjects were in the range of 18–37 years old. The cervical dilation of the subjects during the EMMI mapping was in the range of 3.5 to 9.5 cm. The subjects underwent MRI at ~37 weeks' gestation and then underwent electrical mapping for 1 h during the first stage of labor. Detailed subject information is in Table 1.

### EMMI system

Before MRI scanning, two adhesive measuring tapes were placed on the body surface to guide the placement of MRI markers (Fig. 1). For the abdominal surface, a measuring tape of 30 cm was placed vertically through the midline using the umbilicus as a biomarker and another tape of 61.3 cm was placed horizontally about 4 cm below the fundus over the vertical tape. The fundus was located by an obstetric clinician. Similarly, the back surface, a measuring tape of 30 cm was placed vertically along the spine and ending at the coccyx; another tape of 61.3 cm was placed horizontally at the upper edge of the ilium over the vertical tape. Up to 24 adhesive vinyl-silicon patches containing up to 192 MRI-compatible markers were placed referencing the measuring tapes on the subject's body surface to mimic the electrodes for electrical recordings during active labor. MR images were acquired with the subject in either the supine or left lateral position. MRI was

performed in a 3 T Siemens Prisma or Vida whole-abdomen MRI Scanner with a radial volume interpolated breath-hold examination fast T1-weighted sequence. MR image resolution was 1.56 mm × 1.56 mm, and the slice thickness was 4 mm. BioSemi pin-type unipolar electrodes were assembled into 2 × 4 patches with 3 cm inter-sensor distance. When the subject was in active labor, up to 24 electrode patches containing up to 192 electrodes were placed in the same layout as the MRI markers (Fig. 1). Four ground electrodes were placed at the left and right upper chest, and the left and right lower abdomen, respectively. The body surfaces with electrodes were collected with an optical 3D scanner (Artec 3D, Eva) with structured white light (white 12 LED array and flashbulb) (Fig. 1). The 3D accuracy of the optical scanner is 0.1 mm, and the resolution is 0.5 mm. The subjects' back surfaces were scanned while they were sitting, and their body surfaces were scanned while they were in Fowler's position. After the 3D scanning, the multichannel electrodes were connected to an analog-to-digital converter, and the bioelectricity signals were simultaneously recorded using ActiView Software 8.09 with a 2048 Hz sampling rate.

### EMMI data processing

The MR images, optical 3D scanned surface images, and body surface EMGs were processed using Matlab 2019b (Fig. 1). The geometry, electromyography, uterine electrical activation isochrone map, and early activation map data generated in this study are provided in the Source Data file. The data processing has four parts: (1) Signal pre-processing removes noise and baseline drift in the EMG recordings, eliminates poor contact signals, and excludes motion artifacts; (2) Body-uterus geometry construction delineates the coordinates of body and uterine surface sites and models the triangular meshes to represent these surfaces; (3) Inverse computation uses the Method of fundamental solutions to combine the body-uterus geometry and body surface potentials and compute the uterine surface potentials; (4) Data visualization and analysis is used to post-process the uterine surface potentials. Detailed descriptions of the four components are provided in the following subsections.

The raw electrical recording data were first filtered by a low-pass filter with a cutoff of 40 Hz and then were down-sampled with a mean filter by 20 (sampling rate from 2048 Hz to 102.4 Hz) to reduce the data size and computation time of EMMI. After that, the signals were filtered by a fourth-order Butterworth high-pass filter with a cutoff frequency of 0.34 Hz. The 0.34 Hz high-pass filter aims to reduce the respiration artifacts[2], which otherwise will affect the accuracy of identifying the onset of the EMG burst. Next, an eighth-order

**Table 1 | Subject information**

| Subject number | Delivery history | Age (year) | Labor type | Gestation (week, day) | Membrane status | Cervical dilation range (cm) | Time to full dilation (h) | Birth weight (g) | Body mass index (kg m⁻²) |
|---|---|---|---|---|---|---|---|---|---|
| 1 | Nulliparous | 26 ~ 29 | Spontaneous | 39w5d | Ruptured (Spontaneous) | 3.5 ~ 4 | 7.01 | 3100 | 25.8 |
| 2 | Nulliparous | 30 ~ 33 | Induction | 40w3d | Ruptured (Artificial) | 4 ~ 4 | 4.07 | 3620 | 32.8 |
| 3 | Nulliparous | 18 ~ 21 | Spontaneous | 39w2d | Intact | 5 ~ 5 | 14.41 | 3520 | 33.3 |
| 4 | Nulliparous | 18 ~ 21 | Spontaneous | 39w1d | Ruptured (Spontaneous) | 6.5 ~ 9.5 | 0.23 | 2620 | 22.0 |
| 5 | Nulliparous | 30 ~ 33 | Spontaneous | 39w0d | Ruptured (Artificial) | 7 ~ 9 | 0.77 | 3190 | 29.1 |
| 6 | Multiparous | 30 ~ 33 | Induction | 40w5d | Intact | 4 ~ 4 | 6.95 | 4370 | 23.7 |
| 7 | Multiparous | 34 ~ 37 | Induction | 40w1d | Ruptured (Artificial) | 4 ~ 4.5 | 1.62 | 2980 | 24.6 |
| 8 | Multiparous | 22 ~ 25 | Spontaneous | 38w1d | Intact | 5 ~ 5 | 3.47 | 2865 | 23.0 |
| 9 | Multiparous | 18 ~ 21 | Induction | 37w0d | Ruptured (Artificial) | 5 ~ 6.5 | 1.42 | 3310 | 30.7 |
| 10 | Multiparous | 34 ~ 37 | Spontaneous | 40w6d | Intact/Ruptured (Artificial) * | 6.5 ~ 8 | 0.57 | 3970 | 28.5 |

Cervical dilation range best estimates the cervical dilations at the start and end of electrical recording based on clinical reports. Then these data are subjected to interpolation when the dilation check is more than 2 hours before the beginning or after the end of the study or the cervical dilation changes faster than 1 cm per hour across the electrical recording. *Subject #10 was ruptured artificially ~30 minutes into the electrical recording.

Butterworth low-pass filter with a cutoff frequency of 1 Hz was applied. Finally, a multi-step artifacts detection algorithm was applied to the band-passed signal to detect invalid EMGs containing abnormally large EMG data and distorted body surface potential maps (BSPM). The first quartile of the mean absolute magnitudes of each processed EMG was a reference; any EMG with an absolute magnitude larger than 100 times the reference is detected as an invalid EMG. The median values of the mean absolute magnitude of all ASPMs were a reference, and any ASPMs with mean absolute magnitudes larger than ten times the reference are detected as distorted ASPMs. The local peaks are defined as the peak-peak magnitude within a moving window of 2 seconds. The local artifacts are identified as local peaks with a magnitude larger than ten times the median of the local peaks. An EMG with >50% of the signal contaminated by the local artifacts was defined as an invalid EMG. A BSPM with >50% of the sites contaminated by local artifacts was identified as a distorted BSPM. The invalid EMGs and distorted BSPMs were not included in the following analysis.

The sagittal slices of MR images and optical 3D scans are used to generate the body-uterus geometry. There are three main steps: First, obtain the triangulated mesh surfaces from MR images and the optical 3D scans separately using Amira Software 6.4 and Artec Studio 12. Second, align the optical 3D body surface to the MRI-derived surface and register the optical 3D electrode locations onto the MRI-derived surface (see Supplementary Fig. 1 for details). Third, obtain the coordinates of electrodes and the points on the uterine mesh surface to construct the body-uterine geometry.

The inverse computation combines the body surface electrical potential maps and subject-specific body-uterus geometry to reconstruct uterine potentials. The volume between the uterine surface and body surface $\Omega$ is assumed to be homogeneous, containing no primary electrical source and no eligible inductive effect[13,16,52]. Therefore, the mathematical formulation underlying this inverse problem can be described by the Cauchy problem for Laplace's Eq. (1) with two boundary conditions on the body surface,

$$\nabla^2 \phi(x) = 0, x \in \Omega, \tag{1}$$

Dirichlet condition $\qquad \phi(x) = \phi_{B(x)}, x \in \Gamma_B, \tag{2}$

Neumann condition $\qquad \sigma \dfrac{\partial \phi(x)}{\partial \mathbf{n}} = 0, x \in \Gamma_B. \tag{3}$

$\Gamma_B$ represents the body surface. $\phi_{B(x)}$ is the measured body surface potential at location $x$. $\sigma$ is the conductivity of the volume conductor $\Omega$, which is assumed to be homogeneous. $\mathbf{n}$ is the normal vector on the body surface at $x$. Because the conductivity of air is 0, the right side of Eq. (3) is simplified to 0.

The Method of fundamental solutions[16], a mesh-free method robust to noise, was employed to discretize Eqs. (1–3) and construct the relationship between the measured body surface potentials ($\phi_B$) and uterine surface potentials ($\phi_U$),

$$\Phi_B = \mathbf{A}\Phi_U. \tag{4}$$

$\Phi_B$ is a matrix of M by T, $\Phi_U$ is a matrix of N by T, and $\mathbf{A}$ is a matrix of $M$ by $N$, where $M$ is the number of electrodes on the body surface, $N$ the number of discrete points on the uterine surface, and T the number of potentials maps. The Tikhonov regularization[53] was employed to stabilize the ill-posed inverse computation, which gave a unique solution for each measured abdominal surface potential ($\Phi_U$), and no human intervention was required. A fixed regularization value of 0.01 was used in the Tikhonov-based inverse computation.

After the inverse computation, three types of uterine signals were generated. First, a uterine surface potential map is the electrical potential distribution on the 3D uterine surface at each time point. The time resolution for the uterine potential maps is 102.4 Hz. Second, a uterine electrogram denotes a time series of the electrical potential data at a specific uterine site. Typically, electrograms were calculated at around 320 sites on the uterine mesh. Third, the isochrone map represents the sequence of uterine electrical activation derived from the local activation times during an observation window. The observation window of a UEB started from a time point when the uterus was generally quiet to a time point when the uterus was electrically activated and returned to quiescence.

The activation time for a UEB at each uterine site was defined separately according to the magnitudes of the UEB. The uterine electrogram was first processed with the Teager-Kaiser Energy Operator (TKEO)[54], which improves the signal condition at the onset and offset of the UEB (Supplementary Figs. 2a, b). Then, a root-mean-square (RMS) envelope with a moving window of 7 seconds was derived from the rectified TKEO signal (the black line in Supplementary Fig. 2c), to distinguish signals of activation or baseline. The threshold of the baseline was defined as 1.01 times the median values of the RMS envelope (the blue line in Supplementary Fig. 2c). The threshold for electrical activation was defined as the mean plus twice the standard deviation of the baseline signals. Finally, a UEB was defined as the RMS envelope over the threshold. Considering the low-frequency nature of the uterine, we down-sampled the RMS envelope signals by 20 times when we calculated the activation times. The time resolution for analysis of uterine electrical activation was 5.12 Hz. After the UEB detection, the signal-to-noise ratio (SNR) of the electrogram at each uterine site was defined as the ratio between the power of the UEBs and that of the baseline signals in the Decibel (dB) unit. The uterine electrograms with SNR higher than 5 dB are regarded as qualified contraction signals, and the relevant UEBs are defined as qualified UEBs. The qualified UEBs that last between 5 seconds and 80 seconds are detected as valid electrical activation for labor contractions.

The initiation times of electrical activation were generated from individual uterine surface electrograms at uterine sites, without considering the spatial connectivity. They thus may not preserve spatial continuity in uterine electrical activation pattern. So, with the raw activation time defined on the 3D uterine sites, a series of erosion-dilation operations (similar to morphological filtering) were performed using a customized algorithm written in Matlab to smoothen the isochrone maps: First, tiny activated regions that are wrongly picked up were redefined as inactive by performing an erosion; second, a dilation to the active regions, where erosion sets the activated sites that has one or more inactive neighbors as inactive, and dilation does the opposite. And similarly, we fill tiny inactive regions by performing dilation and then erosion to the active regions, where the new active sites generated by dilation are assigned with activation times by averaging their active neighbors. Because the activation times are relative in isochrone maps, we add or subtract the same number to all activation times so that the first activation starts at 1 second, to avoid potential singularity in calculations. An isochrone map is derived from the updated activation times defined on the 3D mesh of the uterine surface.

The activation curve tracks the percentage of uterine sites that have ever been activated during that contraction. As time increases from zero, the curve starts from the origin, increases by the number of new active uterine sites at the times of new activations, and does not change at the times there are no new activations. The maximal activation ratio (MAR) is the final percentage of the uterine sites that have even been activated. The activation curve slope (ACS) is the slope of the activation process, defined as MAR divided by the time taken to reach MAR, with the unit being percent per second. Fundal early activation ratio (FAR) is the percentage of the number of active fundal sites that activate the earliest 33% across the uterus by the number of fundal sites. The fundus is defined as the uterine region that consists of the nearest neighbors (with Euclidean distance as the metric) of the

anatomical top of the uterus and has one-fourth number of sites of the uterine sites.

## Reporting summary

Further information on research design is available in the Nature Portfolio Reporting Summary linked to this article.

## Data availability

We have declared that the data supporting the findings of this study are available within the paper. The geometry, electromyography, uterine electrical activation isochrone map, and early activation map data generated in this study are provided in the Source Data file. Source data are provided with this paper.

## Code availability

The source codes used to generate EMMI images shown in the figures are provided in the Supplementary Information File.

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

## Acknowledgements

We thank Deborah Frank for editing the manuscript; Jessica Chubiz for managing the subject study; Monica Anderson, Tracy Burger, Emily Diveley, Megan Steiner, Stephanie Pizzella, Cassy Hardy, and Bri Dawson for explaining the study to subjects, obtaining consent, and managing the study; and Nina Punyamurthy, Naomi Goldstein, and Josephine Lau Nga Yue for helping with the initial subject experiments; Rebecca Chen and Hanna Aaron for helping with the initial data preprocessing. This work was supported by the March of Dimes Center Grant (22-FY14-486), by grants from NIH/National Institute of Child Health and Human Development (R01HD094381 to PIs Y. Wang/Cahill; R01HD104822 to PIs Y. Wang/Schwartz/Cahill), by grants from Burroughs Wellcome Fund Preterm Birth Initiative (NGP10119 to PI Y. Wang) and by grants from Bill & Melinda Gates Foundation (INV-005417, INV-035476, and INV-037302 to PI Y. Wang).

## Author contributions

H.W. and Z.W. are co-first authors and contributed equally to the manuscript. H.W. and Y.W. designed the experiments. H.W., Z.W., W.W., Z.S., Z.K.W., Y.L., S.W., H.G., H.X. conducted human experiments. H.W., Z.W., and W.W. segmented the MR images. Q.W. developed and optimized the MRI sequences and scans. P.Z., G.A.M., A.L.S., P.C., and A.G.C. contributed to the study design and guided the clinical studies. H.W., Z.W., and Y.W. developed the EMMI processing pipeline. H.W. and Z.W. processed and analyzed the EMMI data.

## Competing interests

Y.W., A.G.C., P.C., and A.L.S. submitted US Provisional Application No. 62/642,389 titled "System and Method for Noninvasive Electro-myometrial Imaging (EMMI)" for the EMMI technology evaluated in this work. Y.W serves as a scientific consultant for Medtronic and EP Solution and has NIH research funding. The remaining authors declare no competing interests.
