## [Peer Review File · Nature Communications]

Reviewers' Comments:

Reviewer #1:

Remarks to the Author:

The paper by Wang et al, have extended their application of EMMI system from the earlier study in sheep to human application. This system was developed by this group that uses a multimodal approach to provide 3D maps and indices to characterize the uterine activity dynamics in labor.

The results presented from the current study incorporates patient specific geometry through MRI, and multichannel electrical measurements from the abdominal surface to reconstruct the electrical activity across the uterus. The topic and research is very significant and important in terms of understanding of uterine contractile mechanism.

The paper is well written describing the results and methods involved. The concerns remain in the demonstration of the results in a small sample size and the extrapolation of results. Also the practicality of the application in larger population since it involves using MRI for a mapping the uterine geometry.

The introduction touches lots of relevant literature to justify the study and provides a basis for the extension of their sheep EMMI technique to human studies.

The approach involves three steps which are clearly defined which includes MRI recording of the abdominal uterine geometry with fiducial markers, then placing EMG electrodes in the same locations and performing EMG recordings for lasting 15 min with four recordings. The question is that are these recordings back to back or they are sampled at different points of time during the labor process? This question relates to the practicality of the method and also whether it is intended for progress of labor monitoring or more for prediction of term labor or even preterm labor on antepartum period. The authors need to clarify this point.

With regards to comparison with EMG and EMMI, it is true with regards the spatial temporal resolution it is superior but is still requires multiple electrodes (24 patches) which can still be time consuming in when in labor and can have similar drawbacks as EMG.

With regards to comparison with MEG and EMMI, although MEG with comparative spatial resolution requires a shielded room and can be applicable only prior to labor, but EMMI application also requires an MRI in a shielded room so again the practicality in clinical application is limited

All the above questions and comments relate to the fact that from a clinical perspective EMMI does not have an advantage as of yet but definitely can be research tool to understand the mechanism of uterine contractile activity. From a scientific point of view this is an excellent tool and that needs to be emphasized in the discussing rather an imminent clinical application.

The authors in discussion section should discuss potential applications from this perspective scientific research including Pitocin, so that it guide future interventions. The multiparous and nulliparous comparison is in an interesting exploration, and was included in their current study.

This is further between supported by synchrony studies by MMG. Govindan et al *Reprod Sci* 2015 May;22(5):595-601. doi: 10.1177/1933719114556484. Epub 2014 Oct 27. This reference can be used to support the "myometrial memory" observation.

Overall, EMMI is powerful technique to understand the mechanism of uterine contraction and parturition. The paper shows a good transition from a sheep model to a human model. The paper is well-written with good method description. The authors need to focus on the scientific benefits of using EMMI rather than clinical application since as it needs more research since they have limited sample size and recording times.

Reviewer #2:

Remarks to the Author:

Key Results: 1. The human uterus does not express linearly progressive action potential propagation (called here "tissue activation") over long distances; 2. The laboring uterus does not initiate contractions using a pacemaker fixed at a specific location; 3. Some uterine contractions, especially early in the labor process, do not activate all regions of the uterus at the same time; 4. This combination of techniques, while used here to define the physiology of normal labor, may be useful to delineate the physiology of abnormal labor.

Validity: The data were carefully obtained and analyzed and are valid. There is only one question that impacts validity - the definition and use of SYN. Here the authors will very likely be able to

either explain, or easily adapt the manuscript if revised.

Originality and significance: This work is a complex, multimodal study of organ-level physiology of human uterine function during normal term pregnancy. It is a technological tour de force of data collection and analysis to provide meaningful interpretation of term human labor in several clinical presentations. The only two organs of the human body that have not yet established generally accepted mechanisms of action are the brain and the pregnant uterus. This work significantly advances our understanding of the pregnant uterus and is a major contribution to the field. Previous works have provided preliminary results and conclusions that are supported by this work. However, the technology used in the prior works have been underappreciated by clinicians and physiologists for two reasons. First, the technology used to acquire the data was extremely complex (an array of superconducting quantum interference devices that measured magnetic fields). Second, because of the recording restrictions, the subjects were by necessity in early labor and this reduced the general applicability and interpretation of the results. This work uses more accepted and widely-used technology – multichannel electromyography with source location using inverse solutions. Most importantly, this technology can be used to study subjects progressively and in advanced labor. This combination of factors will very likely bridge the gap to clinical acceptance of these results.

Data and methodology. The work reflects excellence in engineering, data collection and methodology. (Although specific questions, comments and suggestions are noted below.) The physiological interpretation of the data is the source of most of the comments described below.

Appropriate use of statistics and treatment of uncertainties. Acceptable, with the exceptions noted as key points that the authors will very likely be able to successfully address (see below, suggested improvements).

Conclusion: The main conclusions are well supported by the data and are important to the field. Several reviewer comments are made below on more tangential conclusions, which will very likely be able to be successfully addressed by the authors.

Suggested improvements: See specific suggestions below. Suggestions labeled “Key Points” must be addressed by the authors. The authors should present the underlying reasons for rejecting any comment labeled “Key Point”. All other suggestions are more minor suggestions of form or grammar and can be accepted or rejected by the authors without comment. Some bullet points presented in the review of the discussion are Key Points.

References: Except for specific suggested additions and corrections (noted below) references adequately support the manuscript.

Clarity and context: Minor comments are included below, many for the sake of clarity and in keeping with traditional use of terminology and approaches in the field.

Inflammatory material: None

Diversity, equity and inclusion: No issues

Detailed comments

Abstract

20 Minor quibble with the word “accurately” – not defined yet

Should emphasize what “entire” means – this is a key point, and should be in the abstract
General – this is a matter of style, but I would encourage the authors to use a brief sentence that notes how many EMG channels, where sensors were distributed, and that an inverse solution is used to localize signal sources. Remove the mention of earlier sheep work.

Introduction

to 36 details why these studies are important to health. While important to many readers, the need in this journal is marginal. Beginning at 36 would be fine.
"motions" is not quite the correct word here. TOCO measures uterine shape change and IUPC measures pressure generated. Simply state these are the two commonly most used techniques clinicians use to assess uterine contractions.
In addition to response to oxytocin, a more important point is that IUPC is invasive, and the non-invasive TOCO cannot identify patients who are in labor, at term or preterm. Here the references from Leveno at Parkland would fit nicely.
New paragraph. "proposed" could/should be "developed"
Why there are two "M"s in EMMI. Too late to change now?
needs to mention the time resolution for performing the images
needs to mention the number of sensors used in sheep
to 54 geometrical deformation is the only pertinent part of this sentence, anticipating clinical challenges needs quite a lot of development if included, which may not be appropriate in the introduction.
Key point. EMMI is probably not "clinical", which, to most readers, implies that it can be used on a daily basis in clinics, by persons requiring minimal additional training. Given the work involved to record 10 subjects, EMMI is not yet a clinical device. This occurs several times in the manuscript and recommend either defining what is meant in this usage, or removing the word altogether when used in this context.

Results

again quibble with the word clinical
please put in the number of EMG sensors/ channels used.
this would be an opportune place to add over time, or sequential images, or something to portray the evolution of contractile activity. How many contractions did you evaluate per subject? Also, you never clearly state that you are or are not using ECG-like sensor (1 cm circle pad, silver chloride with gel).
Key point – Please present a detailed image of the pin-type electrode patches.
...abdomen to guide clinical decisions,
rather than minor adjustments, the locations of the electrode patches were adjusted. To locate the electrode positions on an individual basis, optical 3D...
Key point: these are study subjects, not patients
refer to the Methods section for additional information.
It seems that the geometry of the abdomen and the uterus is less important than the spatial relationship between the abdominal surface and the uterus. If the authors elect to use "geometry", care must be taken in the rest of the manuscript to refer to the abdominal surface or skin rather than just the abdomen.
"instant" is poorly defined, perhaps better to say as a function of time and include an approximate time resolution
Cumbersome and confusing sentence, probably mean to say .. At specific sites on the uterus, we determined the evolution (or the progression) of the uterine electrical activation for each contraction by measuring the start times of the activity derived from the electrograms.
EB is a confusing acronym, would be better to have uterine surface or source included. The whole purpose of this term is to differentiate this signal from the EMG recorded at the skin. "EB" doesn't do that.
probably not "counterparts", rather are the sources of the EMG bursts recorded at the abdominal surface.
simply for clarity and completeness, please label d as left
please identify the subject from Table 1.
Clinical studies although here using EMMI studies would be better since this really isn't a clinical study except for protection of human subjects purposes.
Add IRB approval number.
Did you block enroll or random? How many subjects were studied in total and did you sub-select those to present here. If so how. Over what time frame,
Key point. It looks like the inductions were studied at less cervical dilation – please evaluate this and present results.
Key point. To be complete, it is necessary to include membrane status (intact or ruptured) for each subject.
Key point. The BMI range is limited. Your highest BMI subject is 33.3 and it does not appear

to reflect a random selection of laboring patients at most institutions. Did you select for thin? Do you have abdominal wall thickness measurements – this has great bearing on the precision of the inverse solution and could be mentioned as a reason if you did select thinner subjects. If you did select, did you identify an upper limit for BMI and if so why.

It is OK to use “EMMI” throughout the text, but this heading would read better if you just began with Noninvasive imaging... that is, convert heading to a sentence fragment rather than the declarative statement you use now.
Here, please clearly state your best guess for spatial resolution. Also, not sure what spatial “specificity” is
Patient 2 is at 4 cm and doesn’t change her cervix over the duration of the EMG recordings, hence is in the latent phase of labor. While the information is available in Table 2, it would be good to emphasize this in either the legend or the text.
Key point. This is a very interesting result. Perhaps you will discuss later, but the evolution of the uterine heat map shows much more red in B and D, with the intervening C much cooler. Yet the B and D sections are separated by only one second, then again in E, coolness returns. In Fig 3 the AR does not show these sudden jumps over one second. This rapid oscillation is puzzling since the frequency of the uterine signals is 0.3 to 1 Hz, (call this 0.5 Hz for discussion purposes) which to a first approximation seems to indicate that time resolutions less than 2 seconds is meaningless, and you may be reporting an artifact of the calculations or measurement. Also, the isochrone map in Fig 4 won’t summarize the true activity of a contraction over time if these rapid fluctuations occur. It is possible that I have not understood something, but even if this is the case, please include an explanation for the rapid changes in the results section.
indeed, you defined electrical activation in line 91, thus “we define”.
reading this for the first time gets confusing when you discuss times. The start of an EB is defined as the “activation” in line 92 and 151. Then you call it “activation time” in 157. This may sound picky, but to the first-time reader, it is unclear what time you are referring to – perhaps you are referring to a duration, for example. Perhaps activation “occurs”?
omit “captured”
Regarding the word “regions”. Is this the same use of the term as in reference 39 and 44? Please be explicit on how the AR is calculated.
Key point. Regarding the word “propagation”. Long distance, tissue-level action potential propagation (or activation) is sequential, where going from point A to C requires going through point B. As stated in lines 363-4, your results refute this mechanism. Additionally, recruitment of the entire uterine surface is incomplete, that is, a large portion of the posterior uterus commonly remains inactive but also some anterior uterus remains inactive when the remainder of the anterior uterus is active. These are two different phenomena and when using the phrase “not show clear long-distance propagation” it isn’t clear what you mean.
key point. In the discussion you also state that long-distance propagation is not evident, but you never define the distances that activation occurs. This is essentially how far a tissue-level action potential can travel within the uterine wall (on average). This is an important physiological finding and should go here in the results. Even an approximate value, range, or an upper limit would help define the physiology. Visually, it looks a lot like SARA data from Eswaran – about 8 to 10 cm.
I think you mean “by the end” of the contraction rather than “at the end”. Also this could be defined as the peak, but that is perhaps best left for the discussion section.
to 186 Key point. Here you define SYN, but this is the slope of the development of AR. So, is there negative SYN on the downslope of the contraction profile? (You only present the rise to the contraction peak). SYN as you define it measures the rate of recruitment of regions, which is different from the synchronization of the global uterine contraction. MAR is probably a better representation of uterine synchronization. What you call SYN is more closely related to propagation speed or activation rate, but your term SYN implies uterine synchronization of the regional contractions. Even though you clearly define what it is, please reconsider the term SYN. Also see comments regarding Hill plots and SYN in the discussion.
Key point. This is indeed the clinical TOCO record, but it is not the pressure measurements unless this is an IUPC tracing.

Unrelated to the manuscript – you seem to have not included data through and after the peak of the contraction, and not clearly identified when the peak occurs (at least using TOCO). These data would be highly interesting, since the relationship with EB and peak and offset of the contraction will define a lot of physiology.

In Fig 3 f, the activation curve is sigmoidal. This is highly relevant since it indicates that the evolution of the contraction is mechanism that is regenerative, that is, uses positive feedback. If this is consistent among your subjects (as it appears to be in fig 4), these would be the first clear data confirming a regenerative process in uterine contractility. See additional comments in discussion.

"EMMI imaging" (so I guess you get cash from the ATM machine – "automated teller machine machine") – the I is redundant. Omit either "EMMI" or "imaging"
The current definition of the active phase of labor is beginning at 6 cm dilated. I think you mean experiencing contractions of labor at a certain frequency.
key point. The word "downward" apparently refers to fundal to cervical-going contractions. There are no indications in these data that downward motion, progression, or movement is created by the uterus. This word must be defined and described or removed.
Key point: How is a weak contraction defined with EMMI?
to 230 just the MAR has a higher value, omit "taller" activation curve
again SYN is not synchronization, but the rate of synchronization. It is more likely that this process occurs as detailed in references 39 and 44.
dilation rate was only 0.17, suggesting this subject was in the latent phase of labor.
cumbersome wording, needs reworking
also has cumbersome wording. Unsure of what the authors are trying to convey
Not sure why these FAR values indicate downward contractions. If the fundus is defined as 25% of the total tissue of the uterus, why does 31.25% indicate downward? Since fundal myometrium is thicker than lower uterine segment, the 25% by surface area is probably closer to 30 or 40 % of the tissue volume and number of myocytes. Key point: if the authors wish to discuss "downward", move it to the discussion.
While the cervical maturation is clearly a factor in time to delivery, this explanation for the slow labor progression is a conclusion that should not be in the results section.
applied or used?
– "early-stage" labor is not clinically defined, should rephrase
again the "taller" uterine activation curves are quantified with MAR
I can agree with "suggests", but not "clear". Same comment on early stage. This patient actually is in active labor, possibly could call it "earlier in active labor" or something like that.
Key point. In most subjects in Fig 4 and 5 it appears that the back of the uterus is less active than the front (in 9 of 10 subjects). Is this true, and if so, could this be an artifact of measurement or an artifact of the volume conductor being different anterior/posterior?
again, the definition of active labor is ≥ 6 cm, some of these subjects are still not there, even if they deliver quickly.
one of the subjects is 37.0 weeks, which is not term
quibble with the word significantly – need to have standard deviations, etc. and define p values.
– did the subject report subjectively increasing strength – this would be a significant clinical correlation and if known, should be included.
seemed by whom – EMMI or the subject.
Not sure what electrical maturation is. It is used on line 60, but as a general phrase without definition.
is 1 cm/hr your observed average? – if so give std dev. If from Friedman curve, the 1 cm/hr is not the average, but the dilation rate for 90% of the population.
all the comments on relative cervical softening should be rephrased to simply convey cervical factors rather than a specific characteristic of the cervical exam.
Key point. What is a weak contraction by EMMI?

Discussion

TOCO was defined in line 39 and used throughout the manuscript, no need to repeat
an array of SQUID devices
perhaps would be clearer to the uninitiated if you used the phrase "EMG (also called electrohysterography, EHG)"
If you wish to point out the limitation of SARA, you should also point out the advantage, specifically that MMG signals do not attenuate because of interfaces in the volume conductor.
"all" is perhaps a bit hyperbolic. "Many at different locations" or similar, may be more accurate.
Key point. "highly accurate" is an overstatement of the field. Uterine contraction tracings by EMG are an alternative to TOCO but have a much higher false positive reporting rate (compared with IUPC gold standard). This section of the discussion overstates the success of current EMG-based monitors. It would be more accurate to state that EMG can be used to produce a uterine activity tracing that emulates TOCO and is more reliable than TOCO with obesity, but discussion of the predictive capabilities of EMG should be entered into cautiously since this paper does not deal with any predictions.
reference 39 is erroneously referenced to TOCO (although it should be included elsewhere).
Key point. If you say this, then you have to also say that the propagation "velocity" (really a speed) is controversial and then reference Mikkelsen et al 2013 Acta Obstet Gynecol Scand (plus perhaps a few others, including Catherine Marque, that refute these conclusions). Even though many authors state that they can diagnose true labor and false labor, it would be best to soften all these comments and state that some investigators have suggested that diagnoses can be made using EMG.

334 - 5. Key point. Same, stating that any feature of EMG signals can identify the onset of labor is not true. This ability has never been shown in prospective clinical trials.

it wouldn't be impossible, just not ethical.
high spatial and temporal
not sure why "time domain" is included here and not the prior sentence. What is a counterpart? Is it the source of surface EMG? These 2 sentences are confusing and should be re-written to focus.
One of the strengths of EMMI is that it clearly delineates the myometrial regions that fail to activate during a contraction.
not active within the sensitivity of our recordings. At this point, a discussion as to why the back side of the uterus appears to activate less than the front. Is this a volume conductor effect? See below.
- "and" should be "or" body surface EMG without EMMI or other methods of mapping involving inverse solutions.
new paragraph - Distinct..

361 -3 change "Interestingly" to Our results indicate that ... and is direct evidence against anatomically fixed ... add here [ref 39]

human uterine contraction [add reference 39, 44 and Young, Reproduction 2016 152(2), R51-61]

365 - 6 Not so sure about information obtained over the posterior of the uterus. Suggest you soften this sentence a bit.

"Imaging imaged". One or the other. Actually, the strength of the isochrome map is that incorporates both the timing and extent of activation into a single image. The "entire" aspect has already been emphasized.
- omit "rich"
"times during" would sound better as "over the course of a contraction".
see comments above regarding SYN - as you calculate it , SYN is a rate and doesn't reflect the extent.
Key point: cardiac synchronization is different since all parts of the heart contract with each heartbeat and there is organ-level synchronization by an electrical activation mechanism. You have clearly shown that not all parts of the uterus contract each time and that organ-level synchronization is not by electrical activation (probably is by mechanotransduction, see Young, 2016 Reproduction). There is no reason to define SYN based on cardiac literature. Now that I see why you defined SYN this way, I can now respectfully disagree that this definition adds information to uterine physiology. SYN possibly could be measured as either elapsed time to MAR or simply MAR itself (see comments on analysis of sigmoid curve, below).
and the refractory regions are exactly why understanding the falling phase of the contraction and the time between contractions is important. There is no information on refractory times (or durations, rather) in this work, but since you clearly state "potentially explained", this sentence can stand.
Key point: See comments above on downward. This will require a lot of explaining if you wish to use the word "downward" in the discussion. As it stands, I cannot see that your data supports a downward movement or contraction pattern. The uterus is a pressure-generating organ, it does not use peristalsis or catastalsis to dilate the cervix. This is a common misconception, especially among clinicians and since your data argues against this, this discussion would be a good place to

help stop this misconception.

Where does your data support fundal dominance? Patient 4 is the only subject that shows the fundus activates early in all three contractions. Keep in mind that the fundus is thicker than the lower uterine segment and the number of myocytes per surface area is skewed towards the fundus. Thus, if each myocyte is a potential "pacemaker" for a particular contraction, then your 25% definition of fundal by surface area would likely be the site of origination 30% of the time based purely on probability.
again, the MAR show greater synchronization effects better than SYN, but where does the fundal dominance/ downward statement come from?
the duration of labor is associated with success of vaginal delivery (labor progression) but it is likely not causative. I think you are trying to get at the ability to separate dysfunctional uterine contractility and cervical factors with arrested labor. This discussion section on oxytocin doesn't add a lot, since you didn't give any clinical data on reason for induction, how much oxytocin subjects were on, how long they were on, and the cervix exam at the start of induction. These clinical variables are critical and there are too many to evaluate with only 5 patients in each group (spontaneous versus oxytocin, not multip/nullip). Please reconsider discussing oxytocin in depth.
Myometrial memory is a clever clinical concept, but without any basis in physiology. It is more likely that the cervical compliance of multips is the difference based on restructuring after 10 cm dilation.
how do you define weak versus strong using EMMI? I haven't seen that in your manuscript yet.
"Can be" You haven't previously mentioned any of these complications, and none of your subjects had any of these conditions. If you wish to refer to abnormalities in general, that is fine, but perhaps should avoid proposing clinical trials for specific diseases without justification.

Missing from discussion (each is a key point)

- Sigmoidal curve for activation over time. This addresses the same physiology as the SYN, which I believe is the slope at the inflection point of the curve. These types of curves have been classically addressed using the Hill equation, or similar, and that type of analysis would provide a more comprehensive physiological assessment of the data. The interpretation of the data would also be more relevant to the uterus, rather than forcing a cardiac parameter that relates to organ-level action potential propagation (which does not occur in the uterus)
- In almost all subjects, the back of the uterus, and somewhat the sides, do not appear to be as active as the front. Is this more likely true, or more likely an artifact of measurement? The volume conductor of the back – including muscle/fascia/spine/longer distances is very different than the anterior abdominal wall and may decrease signal amplitudes so that the EMG signals do not rise above threshold in many cases.
- Discussion of MAR as a function of cervical dilation would be good
- Uterine activities appear similar to surface EMG – Was the inverse solution necessary using these multiple pin electrodes? Would this be the same for larger BMI subjects.
- In your early activation maps, is there an "earliest (or first) activation" site you can identify for each contraction. This would be a great help in definitively stating that there is no fixed pacemaker in the uterus.

Missing from the discussion but not a Key Point

- This group has also presented data on sheep. They are uniquely positioned to describe the similarities and differences of the physiology of labor in these two species.

Methods

umbilicus
"near" needs definition. If this is placed by an OB clinician as located at the fundus, state so. If place by a non-obstetrician, state how located and how trained.
coccyx. So where were the tapes placed, using these landmarks?
phrasing typographical error
We assume that the electrodes refer to the EMG sensors/electrodes used later?
"or" or "and" – was only one image obtained (this is how the sentence currently reads, but is confusing). If one image, use of "either" would help.
Fowler was a person and should be capitalized (Butch would have been happy to lend his

name here, but this description is most commonly used in gyn oncology and is not common obstetrical parlance, or to the readers of this journal. Probably semi-recumbent with legs elevated would be better)

While most readers know what an AD-box is, better to write out analog to digital converter and whether it is custom or commercial. I assume the details are given in the sheep publications, so a specific reference would also clear this up.

Key point - Needs to include bandpass for recording, where ground was placed if bipolar, or if monopolar recordings. If not included later, how signal filtering was accomplished, Butterworth, etc., and precise frequencies.

Software

simplify to "processed using Matlab software". The extra words you use are distracting and subject to misinterpretation. What does "custom" refer to - the Matlab software or the software you wrote using Matlab without modifying the Matlab programming.
- "used to determine" or "delineate" the triangular meshes used in inverse calculations. Define is a squishy word.

469 OK, fine to put Butterworth here

Key point. Yikes, why 0.34 to 40? I see you drop to 1 Hz later, but the 40 Hz will capture skeletal muscle. If these frequencies were used in your noise reduction algorithm, please state, otherwise this section is simply confusing. Also, the 0.34 - was this to reduce respiration? Most in the field use 0.2, and some as low as 0.1. The uterine bandpass is so narrow to begin with, narrowing to 0.34 to 1.0 bandpass may lose information. Please justify. However, if you feel this bandpass successfully identifies on/off uterine activity, which is the goal of the paper, then state that clearly and note that others have used slightly wider.
Why down sample here - an anti-alias attempt?
to 477 Rephrase this section. It probably says what you mean to say, but takes three readings to really understand.
"moving" refers to window and is misplaced
How about EMG signal obscured by artifact, rather than bad. Was it exactly half, if so say 50%.
using Thermo
using Artec... to precisely determine electrode locations
to 8 stilted phrasing - reword
change derived to obtained from the MRI and 3D optical scans.
Normal and equal length projections were calculated using a customized algorithm written using Matlab.
not sure what landmarks we are talking about here, clarify.
For the uninitiated reader, please imply why all this was performed, such as to completely define the spatial relationship between the uterus and skin (the medical definition of "abdomen" is not how you use it here)

Inverse computation

to 521 This reviewer is unable to adequately evaluate this section because it is outside his level of expertise. That said, it is important to clearly state if any human intervention was required to optimize or identify the optimal inverse solution, and if the inverse solution always led to a unique result and the algorithm was well-behaved. Clearly state that no human intervention was required, if that was the case.

Data visualization

to 525 key point - need precise time resolution stated here rather than "instant". An approximate value is acceptable here but needs to be defined somewhere.
Key point. The phrase "uterine activation" is a bit of a problem. It appears that the authors mean sequential bioelectrical activity. However, the literature in the field has traditionally used the term "myometrial activation" to mean the conversion of the tissue from quiescent to spontaneously active - a process that takes days or weeks. Simply to avoid this mixing of terminology (which may be long-lasting), perhaps the authors should consider substituting a phrase like, "uterine electrical activation", or "uterine bioelectrical activation".
"EB", or "electrogram burst" was defined on line 92 and while used extensively throughout the manuscript, appears abruptly in the methods section. Purely for readability, it would be best to assign (back at line 92) an abbreviation more closely associated with occurring on the uterus and delineate it from an electrical burst that is observed at the skin. This could be uterine surface signal, source signal, or a variety of other more specific phrases. Seeing EB is mentally translated to electrical burst rather than uterine surface electrogram burst and requires effort to determine the true meaning of the abbreviation each time it appears. While this is a minor point, the definitions described in this manuscript may survive permanently, and care should be taken to assign highly descriptive acronyms when possible.

529. Again the activation problem, probable better to say uterus became electrically active and returned to inactive at that location.

activation time (duration?) or time activation began.
TKEO needs a reference, the Teager energy operator is widely known, but not everyone knows the TKEO or the advantage of this over TEO.
It appears that the 7 second window for RMS sets the time resolution for the sequential images. If this is so, please state here. The 1.01 value to define threshold implies a 1% rise – is this really the case that the noise level was so low.
the baseline statement is either trivial or it is unclear what is meant.

Reading on, 536 to 539 taken as a whole is inconsistent and unclear.

Downsampling may affect temporal resolution and should be noted, also confusion over activation time phrasing comes into play again.
regarded or defined as

547 the EMG is the signal, and I think you are referring to a contraction here. (EMG identified contraction, perhaps)

This is the section that may be considered as requiring human input – or is this purely the result of the algorithm? After reading through this several times, it is possible to figure out what is done. A set up as to why you are doing this could be at the beginning of the paragraph, then point out what problem points do to the images, then the solution you came up with. This will make it much easier to read through.
ever been activated during that contraction – or recently, or some time. Refer to the figures.
SYN looks like the rate of synchronization by your definition – the slope. Please reconsider, and make sure you are saying what you mean to say. For the uterus, synchronization is usually what fraction of the potential sources are active at the same time. Additional comments on SYN are in the discussion.

Extended data figure 1

In general, this figure legend isn't clear. The purpose of the registration is to define the spatial relationships between the uterine surface and the skin. Ideally this would be able to be visualized using only the images, but it is very difficult to do this. While this figure may be technically correct, it could be improved to explain the process more simply.

Here are some specific places

"a" looks like an overlay of green and blue, colors aren't clear. "showing MRI-safe marker locations" (MRI-safe is redundant information for this audience)
Does green represent the results of the procedures that align (or correlate) uterine surface locations with abdominal surface locations?
it isn't clear the difference between rigid and non-rigid.
While the author's choice is technically correct, "orthogonal projection", or perhaps "projected to the nearest location on the skin" would have been more clear than normal, which is commonly used to mean "usual".

Extended data figure 2. This shows processing to the uterine activity (C) of a single channel and perhaps having a summation of channels as (D) would make it look like a TOCO emulation and be more clear. Not sure it is best to use mV on the ordinate for B or C since the energy operator has derivatives in it and not sure that mV adequately represents this.

It looks like getting to the blue boxes are the goal of this figure, but it would help to put an arrow or something that relates back to observing the activation you use in the text.

Key point: The green line does not look like 1% above the baseline as described in the text (1.01, line 536). It looks more like you defined baseline as the maximum value (1.01 times max?)

observed between bursts.

Itemized Author Response Letter (Reviewer#1)

Reviewer #1 (Remarks to the Author):

Q1: The paper by Wang et al, have extended their application of EMMI system from the earlier study in sheep to human application. This system was developed by this group that uses a multimodal approach to provide 3D maps and indices to characterize the uterine activity dynamics in labor. The results presented from the current study incorporates patient specific geometry through MRI, and multichannel electrical measurements from the abdominal surface to reconstruct the electrical activity across the uterus. The topic and research is very significant and important in terms of understanding of uterine contractile mechanism.

Response: We thank the reviewer for the positive comments on the significance of our work.

Q2: The paper is well written describing the results and methods involved. The concerns remain in the demonstration of the results in a small sample size and the extrapolation of results. Also the practicality of the application in larger population since it involves using MRI for a mapping the uterine geometry.

Response: We thank the reviewer for the comments. As pointed out and suggested by the reviewer in the later question below, we added “Although we focus on the technological development and scientific benefits of EMMI system in this work, EMMI holds great potential for wide clinical application.” in this revised manuscript (first sentence of the last 2nd paragraph in Discussion). In a future study, more research with larger sample size and longer recording times will be conducted to further establish EMMI’s clinical usage and applications. In addition, because MRI provides high spatial resolution and accuracy, it is ideal for this first detailed description of human EMMI. A current project conducted in our lab funded by the Gates Foundation aims to develop a wearable, portable EMMI by replacing MRI with an ultrasound scanner in future larger-scale EMMI studies to reduce the cost and increase patient accessibility.

Q3: The introduction touches lots of relevant literature to justify the study and provides a basis for the extension of their sheep EMMI technique to human studies. The approach involves three steps which are clearly defined which includes MRI recording of the abdominal uterine geometry with fiduciary markers, then placing EMG electrodes in the same locations and performing EMG recordings for lasting 15 min with four recordings. The question is that are these recordings back to back or they are sampled at different points of time during the labor process?

Response: We thank the reviewer for the comments. We conducted 4 sessions of 15-minute recordings, which allows break time for the subject to relax, move, use the restroom, etc. Overall, the 4 recording sessions are back-to-back in this study. We will conduct longer recordings with EMMI in future EMMI studies.

Q4: This question relates to the practicality of the method and also whether it is intended for progress of labor monitoring or more for prediction of term labor or even preterm labor on antepartum period. The authors need to clarify this point.

Response: We thank the reviewer for the comments. While establishing the EMMI technique in this project using MRI and commercial electrodes made by BioSemi, we are working to develop low-cost, elastic, printed electrodes [Lo, L., et al., ACS Appl Mater Interfaces. 2021 May 12;13(18): 21693-21702. Lo, L., et al., ACS Appl Mater Interfaces. 2022 Feb. 14;14(7): 9570–9578. Lo, L., et al., ACS Nano. 2022 .] and replace MRI with an ultrasound scanner in EMMI. These efforts are currently funded by the Gates Foundation and are actively conducted in our lab, which will significantly enhance the practicality and accessibility of EMMI.

We agree with the reviewer on future EMMI clinical applications. EMMI will be used in several important clinical applications: (1) Labor progression monitoring: EMMI will be employed to provide novel uterine electrophysiological biomarkers reflecting labor progression. Particularly, EMMI will be used to detect labor arrest and facilitate clinical decision-making. (2) Antepartum mapping: EMMI will be used in patients with preterm contractions during the antepartum period to detect productive preterm contractions that will lead to preterm labor.

Q5: With regards to comparison with EMG and EMMI, it is true with regards the spatial temporal resolution it is superior but is still requires multiple electrodes (24 patches) which can still be time consuming in when in labor and can have similar drawbacks as EMG. With regards to comparison with MEG and EMMI, although MEG with comparative spatial resolution requires a shielded room and can be applicable only prior to labor, but EMMI application also requires an MRI in a shielded room so again the practicality in clinical application is limited

Response: We agree with the reviewer that EMMI is the electrophysiological imaging technique built based on the EMG technique. In order to speed up the electrode application, we developed electrode patches, and we are also developing printed, elastic, disposable electrodes [Lo, L., et al., ACS Appl Mater Interfaces. 2021 May 12;13(18): 21693-21702. Lo, L., et al., ACS Appl Mater Interfaces. 2022 Feb. 14;14(7): 9570–9578.] to further enhance the accessibility of EMMI.

After a short clinical MRI scan, the EMMI will be conducted for an extended period in a regular clinical room during the antepartum period and labor. In comparison, MEG requires the entire recording to be conducted in the MEG room, which is not suitable for labor imaging. We agree with the reviewer that MRI can still be a limiting factor in the scale-up study. Following this initial EMMI feasibility study with MRI, we are developing ultrasound-based EMMI to reduce the cost and increase the accessibility of EMMI.

Q6: All the above questions and comments relate to the fact that from a clinical perspective EMMI does not have an advantage as of yet but definitely can be research tool to understand the mechanism of uterine contractile activity. From a scientific point of view this is an excellent tool and that needs to be emphasized in the discussing rather an imminent clinical application.

Response: We thank the reviewer for the comments. We completely agree with the reviewer, and we focus on the scientific benefits of using the novel EMMI rather than its imminent clinical application in this revised manuscript. Please also refer to the responses to the questions above.

Q7: The authors in discussion section should discuss potential applications from this perspective scientific research including Pitocin, so that it guide future interventions. The multiparous and nulliparous comparison is in an interesting exploration, and was included in their current study.

Response: We thank the reviewer for the comments. We added the following statement in the discussion on the potential EMMI application of Pitocin.

“Pitocin, a synthetic version of the naturally derived oxytocin, is the most commonly used medication to augment labor progress. However, oxytocin is released in a pulsatile fashion, while Pitocin is given in an escalating, continuous drip, titrated to contraction frequency and intensity. But nothing is known regarding the direct effect of Pitocin on myometrial electrical activation and propagation, and whether or not this non-physiologic delivery is optimal has never been explored. EMMI will enable these explorations and uncover further insights regarding the physiology of labor and labor management.” (See Line 441-448).

Q8: This is further between supported by synchrony studies by MMG. Govindan et al Reprod Sci 2015 May;22(5):595-601. doi: 10.1177/1933719114556484. Epub 2014 Oct 27. This reference can be used to support the “myometrial memory” observation.

Response: We thank the reviewer for the comments. We included the literature as part of our discussion as suggested by the reviewer.

Q9. Overall, EMMI is powerful technique to understand the mechanism of uterine contraction and parturition. The paper shows a good transition from a sheep model to a human model. The paper is well-written with good method description. The authors need to focus on the scientific benefits of using EMMI rather than clinical application since as it needs more research since they have limited sample size and recording times.

Response: We thank the reviewer for the comments. We completely agree with the reviewer, and we focus on the scientific benefits of using the novel EMMI rather than its imminent clinical application in this revised manuscript. Please also refer to the responses to the questions above.

Itemized Author Response Letter (Reviewer#2)

Reviewer #2 (Remarks to the Author):

Key Results: 1. The human uterus does not express linearly progressive action potential propagation (called here “tissue activation”) over long distances; 2. The laboring uterus does not initiate contractions using a pacemaker fixed at a specific location; 3. Some uterine contractions, especially early in the labor process, do not activate all regions of the uterus at the same time; 4. This combination of techniques, while used here to define the physiology of normal labor, may be useful to delineate the physiology of abnormal labor.

Response: We thank the reviewer for the very important and positive review.

Validity: The data were carefully obtained and analyzed and are valid. There is only one question that impacts validity - the definition and use of SYN. Here the authors will very likely be able to either explain, or easily adapt the manuscript if revised.

Response: We thank the reviewer for the very important and positive review. We agree with the reviewer and replace SYN with activation curve slope (ACS).

Originality and significance: This work is a complex, multimodal study of organ-level physiology of human uterine function during normal term pregnancy. It is a technological tour de force of data collection and analysis to provide meaningful interpretation of term human labor in several clinical presentations. The only two organs of the human body that have not yet established generally accepted mechanisms of action are the brain and the pregnant uterus. This work significantly advances our understanding of the pregnant uterus and is a major contribution to the field. Previous works have provided preliminary results and conclusions that are supported by this work. However, the technology used in the prior works have been underappreciated by clinicians and physiologists for two reasons. First, the technology used to acquire the data was extremely complex (an array of superconducting quantum interference devices that measured magnetic fields). Second, because of the recording restrictions, the subjects were by necessity in early labor and this reduced the general applicability and interpretation of the results. This work uses more accepted and widely-used technology – multichannel electromyography with source location using inverse solutions. Most importantly, this technology can be used to study subjects progressively and in advanced labor. This combination of factors will very likely bridge the gap to clinical acceptance of these results.

Response: We thank the reviewer for the very important and positive review.

Data and methodology. The work reflects excellence in engineering, data collection and methodology. (Although specific questions, comments and suggestions are noted below.) The physiological interpretation of the data is the source of most of the comments described below.

Response: We thank the reviewer for the very important and positive review. We made all required revisions to address the raised questions (see details below).

Appropriate use of statistics and treatment of uncertainties. Acceptable, with the exceptions noted as key points that the authors will very likely be able to successfully address (see below, suggested improvements).

Response: We thank the reviewer for the very important and positive review. We made all required revisions to address the raised questions (see details below).

Conclusion: The main conclusions are well supported by the data and are important to the field. Several reviewer comments are made below on more tangential conclusions, which will very likely be able to be successfully addressed by the authors.

Response: We thank the reviewer for the very important and positive review. We made all required revisions to address the raised questions (see details below).

Suggested improvements: See specific suggestions below. Suggestions labeled “Key Points” must be addressed by the authors. The authors should present the underlying reasons for rejecting any comment labeled “Key Point”. All other suggestions are more minor suggestions of form or grammar and can be accepted or rejected by the authors without comment. Some bullet points presented in the review of the discussion are Key Points.

Response: We thank the reviewer for the very important and positive review. We made all required revisions to address the raised questions (see details below).

References: Except for specific suggested additions and corrections (noted below) references adequately support the manuscript.

Response: We thank the reviewer for the very important and positive review. We made all required revisions to address the raised questions (see details below).

Clarity and context: Minor comments are included below, many for the sake of clarity and in keeping with traditional use of terminology and approaches in the field.

Response: We thank the reviewer for the very important and positive review. We made all required revisions to address the raised questions (see details below).

Inflammatory material: None

Diversity, equity and inclusion: No issues

Detailed comments

Abstract

20 Minor quibble with the word “accurately” – not defined yet

Response: We thank the reviewer for the comment. In the sheep EMMI study (Wu, W., et al., *Sci Transl Med*, 2019. 11(483)), we conducted a head-to-head comparison between EMMI imaged uterine electrical signals and directly measured uterine electrical activity to demonstrate the accuracy of EMMI.

**Key point:** Should emphasize what “entire” means – this is a key point, and should be in the abstract General – this is a matter of style, but I would encourage the authors to use a brief sentence that notes how many EMG channels, where sensors were distributed, and that an inverse solution is used to localize signal sources. Remove the mention of earlier sheep work.

Response: We thank the reviewer for the comment. “Entire” means that EMMI can reconstruct uterine surface electrical activity at any area of the uterine surface. In comparison, invasive measurement usually can only cover part of the uterus due to the limited accessibility of placing electrodes directly on the uterine surface during the surgery. We added the electrode/sensor information in the abstract.

“to generate noninvasively the uterine surface electrical potential maps, electrograms, and activation sequence through an inverse solution using up to 192 electrodes distributed around the abdomen surface”. (See Line 26-28).

Introduction

to 36 details why these studies are important to health. While important to many readers, the need in this journal is marginal. Beginning at 36 would be fine.

Response: We thank the reviewer for the comment. Considering the potential readers from different fields, we prefer to keep the background information within the text limit of the journal.

“motions” is not quite the correct word here. TOCO measures uterine shape change and IUPC measures pressure generated. Simply state these are the two commonly most used techniques clinicians use to assess uterine contractions.

Response: We thank the reviewer for the comment and removed the word “motion” in the revised manuscript.

In addition to response to oxytocin, a more important point is that IUFC is invasive, and the non-invasive TOCO cannot identify patients who are in labor, at term or preterm. Here the references from Leveno at Parkland would fit nicely.

Response: We thank the reviewer for the advice and incorporate the suggestion and add the reference. “Particularly, previous studies found that TOCO cannot identify patients who are in labor, at term or preterm. [Iams, Jay D., et al. N Engl J Med. 2002 Jan 24;346(4):250-5. Chao, Tamara T., et al. Obstet Gynecol. 2011 Dec;118(6):1301-1308.]”. (See Line 44).

New paragraph. “proposed” could/should be “developed”

Response: We thank the reviewer for the advice and incorporate the suggestion in the revised manuscript.

Why there are two “M”s in EMMI. Too late to change now?

Response: The term is Electromyometrial Imaging (EMMI).

needs to mention the time resolution for performing the images

Response: We add “up to 2kHz” after temporal resolution in the revised manuscript. (See Line 51).

needs to mention the number of sensors used in sheep

Response: We added the number of electrodes (up to 192 electrodes) in the revised manuscript.

to 54 geometrical deformation is the only pertinent part of this sentence, anticipating clinical challenges needs quite a lot of development if included, which may not be appropriate in the introduction.

Response: We agree that clinical challenges will require more development and we limit ourselves in the electrical noise contamination in the revised manuscript.

**Key point.** EMMI is probably not “clinical”, which, to most readers, implies that it can be used on a daily basis in clinics, by persons requiring minimal additional training. Given the work involved to record 10 subjects, EMMI is not yet a clinical device. This occurs several times in the manuscript and recommend either defining what is meant in this usage, or removing the word altogether when used in this context.

Response: We agree with the reviewer and replaced “clinical EMMI” with “human EMMI” thorough the revised manuscript.

Results

again quibble with the word clinical

Response: We agree with the reviewer and replaced “clinical EMMI” with “human EMMI” thorough the revised manuscript.

please put in the number of EMG sensors/ channels used.

Response: We added the number of EMG sensors in the revised manuscript. (See Line 71).

this would be an opportune place to add over time, or sequential images, or something to portray the evolution of contractile activity. How many contractions did you evaluate per subject? Also, you never clearly state that you are or are not using ECG-like sensor (1 cm circle pad, silver chloride with gel).

Response: We add the “sequential frames” in “electrical activities in sequential frames across the 3D uterus.” (See Line 72).

The number of contractions varies dramatically in different subjects in this study, ranging from 3 to 21. We also added in the method section that we are using BioSemi “pin-type electrodes”, which have been used to measure ECG and EMG.

**Key point** – Please present a detailed image of the pin-type electrode patches.

Response: We added a detailed image of an electrode patch in Fig. 1c.

73 ...abdomen to guide clinical decisions,

Response: We revised as the reviewer suggested.

rather than minor adjustments, the locations of the electrode patches were adjusted. To locate the electrode positions on an individual basis, optical 3D...

Response: We revised as the reviewer suggested.

**Key point:** these are study subjects, not patients

Response: We revised as the reviewer suggested by replacing “patients” with “subjects” throughout the manuscripts.

refer to the Methods section for additional information.

Response: We revised as the reviewer suggested.

It seems that the geometry of the abdomen and the uterus is less important than the spatial relationship between the abdominal surface and the uterus. If the authors elect to use “geometry”, care must be taken in the rest of the manuscript to refer to the abdominal surface or skin rather than just the abdomen.’

Response: We agree with the reviewer that we should use the “body surface” instead of the “abdomen”. We made the changes throughout the revised manuscript.

“instant” is poorly defined, perhaps better to say as a function of time and include an approximate time resolution

Response: We agree with the reviewer, and we changed it to “on the uterine surface as a function of time every 10 milliseconds”. (See Line 94).

Cumbersome and confusing sentence, probably mean to say .. At specific sites on the uterus, we determined the evolution (or the progression) of the uterine electrical activation for each contraction by measuring the start times of the activity derived from the electrograms.

Response: We agree with the reviewer and revised accordingly. “During a contraction, we determine the uterine electrical activation times by measuring the start times of uterine electrogram burst (UEB) at each uterine surface site, which will be used to form the isochrone map (Fig. 1h)”. (See Line 96-99).

EB is a confusing acronym, would be better to have uterine surface or source included. The whole purpose of this term is to differentiate this signal from the EMG recorded at the skin. “EB” doesn’t do that.

Response: We completely agree with the reviewer, and we use “uterine electrical burst (UEB)” throughout the revised manuscript.

probably not “counterparts”, rather are the sources of the EMG bursts recorded at the abdominal surface.

Response: We agree with the reviewer and deleted the sentence to avoid confusion.

simply for clarity and completeness, please label d as left

Response: We labeled “left view” in panel d.

please identify the subject from Table 1.

Response: The data in Figure 1 was from a feasibility testing subject enrolled during the early development phase of the human EMMI system. Limited electrical recordings (~10mins) were acquired for this patient, and the subject is not included in Table 1.

Clinical studies although here using EMMI studies would be better since this really isn't a clinical study except for protection of human subjects purposes.

Response: We change “clinical implementation” to “study implementation”. (See Line 120).

Add IRB approval number.

Response: We add IRB #201612140. (See Line 121).

Did you block enroll or random? How many subjects were studied in total and did you sub-select those to present here. If so how. Over what time frame,

Response: We utilize random enrollment. Between Aug. 2017 and Aug. 2021, a total of 10 subjects (reported in this manuscript) completed the entire EMMI process and passed the quality control on MRI and electrical recording. (Of note, labor and delivery research operations were suspended from March 2020 until March 2021 due to the SARS-CoV-2 pandemic.)

**Key point.** It looks like the inductions were studied at less cervical dilation – please evaluate this and present results.

Response: Since the induction was usually arranged, we can get the EMMI system ready (prepare electrodes with gel, etc.) for the study ahead of time. That is why in our cohort, the induction subjects were imaged by EMMI usually at less cervical dilation. In comparison, the EMMI team was notified after the subjects with spontaneous labor had already gotten into the labor-delivery unit. When the EMMI system was ready and moved to the delivery room for the study, the cervix dilation was usually larger than in the induction group.

**Key point.** To be complete, it is necessary to include membrane status (intact or ruptured) for each subject.

Response: We thank the reviewer for the suggestion, and we added the membrane status to the revised Table 1.

**Key point.** The BMI range is limited. Your highest BMI subject is 33.3 and it does not appear to reflect a random selection of laboring patients at most institutions. Did you select for thin? Do you have abdominal wall thickness measurements – this has great bearing on the precision of the inverse solution and could be mentioned as a reason if you did select thinner subjects. If you did select, did you identify an upper limit for BMI and if so why.

Response: In this first human EMMI study using a clinical MRI scanner, one of our inclusion criteria is that the pre-pregnancy BMI is not bigger than 40 to ensure pregnant patients would comfortably fit inside the MRI scanner at around 37 weeks. We selected the Siemens 3T Vida scanner for this project because it has the largest bore size (70 cm) available in our institute and is ideal for pelvic imaging in pregnant women.

It is OK to use “EMMI” throughout the text, but this heading would read better if you just began with Noninvasive imaging... that is, convert heading to a sentence fragment rather than the declarative statement you use now.

Response: We thank the reviewer for the suggestion, and we reworded the heading as suggested. (See Line 134).

Here, please clearly state your best guess for spatial resolution. Also, not sure what spatial “specificity” is.

Response: We thank the reviewer for the suggestion, and we replaced “specificity” with “resolution”. Currently, the uterine surfaces consist of 320 vertices evenly and we estimate the spatial resolution is 2.5 cm.

Patient 2 is at 4 cm and doesn’t change her cervix over the duration of the EMG recordings, hence is in the latent phase of labor. While the information is available in Table 2, it would be good to emphasize this in either the legend or the text.

Response: We thank the reviewer for the suggestion, and we added “in the latent phase of labor” in the legend as suggested. (See Line 152).

**Key point.** This is a very interesting result. Perhaps you will discuss later, but the evolution of the uterine heat map shows much more red in B and D, with the intervening C much cooler. Yet the B and D sections are separated by only one second, then again in E, coolness returns. In Fig 3 the AR does not show these sudden jumps over one second. This rapid oscillation is puzzling since the frequency of the uterine signals is 0.3 to 1 Hz, (call this 0.5 Hz for discussion purposes) which to a first approximation

seems to indicate that time resolutions less than 2 seconds is meaningless, and you may be reporting an artifact of the calculations or measurement. Also, the isochrone map in Fig 4 won't summarize the true activity of a contraction over time if these rapid fluctuations occur. It is possible that I have not understood something, but even if this is the case, please include an explanation for the rapid changes in the results section.

Response: We thank the reviewer for the comments. As well known, one uterine electrical burst (UEB) is composed of many electrical spikes. The potential maps shown in Fig. 3 demonstrated the spatial distribution of uterine surface potentials during several spikes within a uterine contraction burst. As the reviewer observed, the uterine potential map shown from panels c to d represents the evolution from a negative peak to a positive peak of spikes. This rapid oscillation pattern reflects the uterine spatial-temporal behavior at the spike scale (~0.6-1Hz). In EMMI, the contraction activation time is defined as the start time of a uterine surface electrogram burst, which explains why isochrone maps will not reflect the complex potential map patterns on the scale of spikes. Our data suggested that the spatial-temporal dynamics of the spike within the uterine electrical burst (UEB) also carry rich physiology and pathological information and will be worthwhile to be elucidated in future dedicated research beyond this work. To clarify, we add the following statement in the result section: "The 4D spatial-temporal uterine surface potential map imaged by EMMI (in Fig. 2b through 2e) reveals the evolution of phase and magnitude inside a UEB". (See Line 156).

In addition, our EMMI system acquired the EMG at a 2kHz sampling rate, which is redundant for uterine EMG analysis. Therefore, we first conduct a 40Hz low pass filter to prepare the EMG signals for the down-sampling of the raw signal to 100Hz. Then we passed the EMG to a band-pass filter 0.34-1Hz. The 0.34 Hz low cut-off frequency is to remove the respiration artifacts, which is less than 0.34 Hz. The 0.34 - 1Hz is used in the field and has been reported to reveal the main features of uterine contractions [Vasak B, et. al. Am J Obstet Gynecol. 2013 Sep;209(3):232.e1-8]. The resulting EMG signals are then used for EMMI reconstruction to generate uterine surface potential signals (0.34-1Hz) with a 100Hz sampling rate. Therefore, as the reviewer pointed out, the uterine surface EMG will not have frequency components faster than 1Hz. Based on Nyquist-Shannon sampling theorem, we need at least a 2 Hz sampling rate to capture all the frequency components contained in our EMMI uterine surface EMG. However, we found that the uterine surface potential maps demonstrated more refined temporal details due to the phase/magnitude difference across the uterine EMG signals (as shown in Panel b-e in Fig. 3). Therefore, the selection of 100Hz sampling rate is employed to preserve the complete spectrum information, and also faithfully reflect the detailed uterine potential map evolution during contractions.

indeed, you defined electrical activation in line 91, thus "we define".

Response: We reworded the sentence based on the reviewer's suggestion. (See Line 97 and Line 161).

reading this for the first time gets confusing when you discuss times. The start of an EB is defined as the "activation" in line 92 and 151. Then you call it "activation time" in 157. This may sound picky, but to the first-time reader, it is unclear what time you are referring to – perhaps you are referring to a duration, for example. Perhaps activation "occurs"?

Response: Based on the reviewer's comment, we replaced "electrical burst (EB)" with "uterine electrical burst (UEB)" and added the definition of activation time as in the revised manuscript: "The term 'uterine electrical activation time' or just 'activation time' is used here to refer to the initiation time of the UEB." (See Line 162).

omit "captured"

Response: We reworded the sentence based on the reviewer's suggestion.

Regarding the word "regions". Is this the same use of the term as in reference 39 and 44? Please be explicit on how the AR is calculated.

Response: The region here refers to the uterine area where the myometrium was activated at the time labeled in Fig. 3c. We added "Activation ratio (AR) is calculated as dividing the area of the activated uterine region by the total uterine area as a function of time." (See Line 183).

**Key point.** Regarding the word "propagation". Long distance, tissue-level action potential propagation (or activation) is sequential, where going from point A to C requires going through point B. As stated in lines 363-4, your results refute this mechanism. Additionally, recruitment of the entire uterine surface is incomplete, that is, a large portion of the posterior uterus commonly remains inactive but also some anterior uterus remains inactive when the remainder of the anterior uterus is active. These are two different phenomena and when using the phrase "not show clear long-distance propagation" it isn't clear what you mean.

Response: We agree with the reviewer that these are both different phenomena observed by EMMI. We explicitly point this out in the revised manuscript.

"First, EMMI can detect the region that is active or inactive during a contraction. When a large portion of the uterus remains inactive, there is insufficient myometrium to support long-distance propagation. Second, even when there is a large portion of the uterus that is active, we did not find the cardiac-like long-distance propagation within the activated region." (See Line 187-191).

**key point.** In the discussion you also state that long-distance propagation is not evident, but you never define the distances that activation occurs. This is essentially how far a tissue-level action potential can travel within the uterine wall (on average). This is an important physiological finding and should go here in the results. Even an approximate value, range, or an upper limit would help define the physiology. Visually, it looks a lot like SARA data from Eswaran – about 8 to 10 cm.

Response: We agree with the reviewer that we did not observe a smooth continuous long-distance propagation. Instead, the uterine activation pattern is a very complicated pattern on the 3D surface, as demonstrated by the isochrone map. Based on the manual measurement, we estimate that the longest

propagation distance (uterine surface distance) observed in our study is 8-14 cm on the 3D uterine surface. We will continue to confirm this finding in the future larger cohort.

I think you mean “by the end” of the contraction rather than “at the end”. Also this could be defined as the peak, but that is perhaps best left for the discussion section.

Response: We agree with the reviewer and modified the definition of MAR as follows: “Maximal activation ratio (MAR) can be quantified as the percentage of the total activated myometrium by the end of the contraction.” (See Line 195).

Key point. 184 to 186 Here you define SYN, but this is the slope of the development of AR. So, is there negative SYN on the downslope of the contraction profile? (You only present the rise to the contraction peak). SYN as you define it measures the rate of recruitment of regions, which is different from the synchronization of the global uterine contraction. MAR is probably a better representation of uterine synchronization. What you call SYN is more closely related to propagation speed or activation rate, but your term SYN implies uterine synchronization of the regional contractions. Even though you clearly define what it is, please reconsider the term SYN. Also see comments regarding Hill plots and SYN in the discussion.

Response: We agree the current definition of SYN is the slope of the development of the AR curve. Since we are focusing on the activation phase of the uterine contraction in this study, we only derive the upslope of the AR curve. Similarly, we can also derive the downslope part of the AR curve, potentially reflecting the recovery/repolarization process of the myometrium. The downslope of the AR curve will lead to a negative slope.

We agree with the reviewer that the current definition of SYN as a uterine synchronization index may be improper. Based on the reviewer’s suggestion, we now use the activation curve slope (ACS) to replace SYN in the revised manuscript.

We sincerely thank the reviewer for pointing us to the Hill equation/plots, and we agree that we can potentially develop better uterine physiology-related EMMI indices by fitting the sigmoid activation curve with Hill equations/plots in our future work.

**Key point.** This is indeed the clinical TOCO record, but it is not the pressure measurements unless this is an IUPC tracing. Unrelated to the manuscript – you seem to have not included data through and after the peak of the contraction, and not clearly identified when the peak occurs (at least using TOCO). These data would be highly interesting, since the relationship with EB and peak and offset of the contraction will define a lot of physiology.

Response: We completely agree with the comments on the TOCO signal and will change the title of Fig. 3a. For this particular clinical TOCO record, the TOCO signal was saturated to the maximal values, which did not show the peak of the signal. We will better work with the clinical team in future studies to avoid such TOCO signal saturation by adjusting the TOCO system in a timely manner.

In Fig 3 f, the activation curve is sigmoidal. This is highly relevant since it indicates that the evolution of the contraction is mechanism that is regenerative, that is, uses positive feedback. If this is consistent among your subjects (as it appears to be in fig 4), these would be the first clear data confirming a regenerative process in uterine contractility. See additional comments in discussion.

Response: We completely agree with the comments and thank the reviewer for pointing us to the Hill equation.

“EMMI imaging” (so I guess you get cash from the ATM machine – “automated teller machine machine”) – the I is redundant. Omit either “EMMI” or “imaging”

Response: We agree with the reviewer and removed the “imaging” after EMMI. We revised the manuscript to avoid redundancy. (See Line 223).

The current definition of the active phase of labor is beginning at 6 cm dilated. I think you mean experiencing contractions of labor at a certain frequency.

Response: We thank the reviewer for the advice, and we changed the section title from “EMMI imaging of nulliparous patients during the active phase of term labor” to “Imaging uterine contraction during labor in nulliparous patients”. Similar changes are made to the next section title. (See Line 223).

**key point.** The word “downward” apparently refers to fundal to cervical-going contractions. There are no indications in these data that downward motion, progression, or movement is created by the uterus. This word must be defined and described or removed.

Response: We agree with the reviewer and remove the mechanical term “downwards”.

**Key point:** How is a weak contraction defined with EMMI?

Response: We agree with the reviewer and by “weak” we refer to low MAR. We changed the sentence to “the uterine contractions involve a small amount of myometrium, as indicated by the low MAR, at the early stage of active labor”. (See Line 236).

to 230 just the MAR has a higher value, omit “taller” activation curve

Response: We made changes based on the reviewer’s suggestion.

again SYN is not synchronization, but the rate of synchronization. It is more likely that this process occurs as detailed in references 39 and 44.

Response: We made changes based on the reviewer's suggestion above and used ACS to replace SYN in the revised manuscript.

dilation rate was only 0.17, suggesting this subject was in the latent phase of labor.

Response: We include "this subject was in the latent phase of labor". (See Line 256).

cumbersome wording, needs reworking

Response: We follow the reviewer's suggestion and rework the sentence as "Similarly, we did not observe the identical activation patterns and existence of consistent initiation sites in Subject #2." (See Line 257-258).

also has cumbersome wording. Unsure of what the authors are trying to convey

Response: We follow the reviewer's suggestion and rework the sentence as "It was also noted that the early activation region (red-yellow) was dominantly located in the fundus region for the three contractions (33.75%, 46.25%, and 31.25%)." (See Line 261-262).

Not sure why these FAR values indicate downward contractions. If the fundus is defined as 25% of the total tissue of the uterus, why does 31.25% indicate downward? Since fundal myometrium is thicker than lower uterine segment, the 25% by surface area is probably closer to 30 or 40 % of the tissue volume and number of myocytes. Key point: if the authors wish to discuss "downward", move it to the discussion.

Response: We agree with the reviewer's comments. FAR reflects how many percentages of the early activation region (defined as the early 1/3 of the active uterine surface area) are in the fundus region. We removed the description of mechanical deformation "downward", we also have the corresponding responses in the discussion.

While the cervical maturation is clearly a factor in time to delivery, this explanation for the slow labor progression is a conclusion that should not be in the results section.

Response: We follow the reviewer's suggestion and minimize the discussion contained in the results section and move them to the discussion session. (See Line 411-412).

applied or used?

Response: We changed "applied" to "used". (See Line 266).

– “early-stage” labor is not clinically defined, should rephrase

Response: We changed “which was much faster than the early-stage” to “which was much faster than that observed in Patient #1-3”. (See Line 268).

again the “taller” uterine activation curves are quantified with MAR

Response: We changed “the uterine activation curves are much taller.” to “the MAR values of those contractions are much higher”. (See Line 271).

I can agree with “suggests”, but not “clear”. Same comment on early stage. This patient actually is in active labor, possibly could call it “earlier in active labor” or something like that.

Response: We removed “clear” and used “earlier in active labor”. (See Line 275).

**Key point.** In most subjects in Fig 4 and 5 it appears that the back of the uterus is less active than the front (in 9 of 10 subjects). Is this true, and if so, could this be an artifact of measurement or an artifact of the volume conductor being different anterior/posterior?

Response: We thank the reviewer for the comments. We also have the same observation in our subjects with good quality MRI and electrical recordings. Therefore, we believe the results are true based on the best performance of our current version of the EMMI system. However, we also agree that the distance between the lower back and posterior uterus is longer than the distance between the anterior abdomen and anterior uterus, which creates asymmetric geometry of the volume conductor delineated by the uterus and body surfaces and leads to the lower electrical signal SNR on the back. This could make the conventional fixed penalty term selection in the Tikhonov inverse computation less optimal, especially for the inverse computation of the potentials on the posterior uterine wall. A better inverse algorithm with an unfixed penalty term will help to further confirm our findings in future development.

again, the definition of active labor is ≥ 6 cm, some of these subjects are still not there, even if they deliver quickly.

Response: We thank the reviewer for the advice, and we changed the section title from “EMMI imaging of multiparous patients during the active phase of term labor” to “Imaging uterine contractions during labor in multiparous patients”. (See Line 286).

one of the subjects is 37.0 weeks, which is not term

Response: We thank the reviewer for the comment. We describe preterm labor as labor that begins early, before 37 weeks of pregnancy, and labor at and after 37 0/7 weeks are grouped in our term group. Since 37 0/7 weeks can be treated as “early term”, we removed the “term” from the section title.

quibble with the word significantly – need to have standard deviations, etc. and define p values.

Response: We thank the reviewer for the comments, and we removed the word 'significantly' from the statement.

– did the subject report subjectively increasing strength – this would be a significant clinical correlation and if known, should be included.

Response: We thank the reviewer for the comments. Unfortunately, we do not have clinical records of the subject's self-report of contraction strength.

seemed by whom – EMMI or the subject.

Response: We thank the reviewer for the comments, and we reform the sentence to "In the early stage of active labor, EMMI found that the uterine contraction in multiparous patients is stronger than those in nulliparous patients, which may suggest earlier electrical maturation." (See Line 299).

Not sure what electrical maturation is. It is used on line 60, but as a general phrase without definition.

Response: We thank the reviewer for the comment. We added, "which is the development of the capacity of the uterus to appropriately initiate and conduct electrical signals across the myometrium". (See Line 64-66).

is 1 cm/hr your observed average? – if so give std dev. If from Friedman curve, the 1 cm/hr is not the average, but the dilation rate for 90% of the population.

Response: We agree with the reviewer and add "for 90% of the population" after "the average rate of 1 cm per h". (See Line 308).

all the comments on relative cervical softening should be rephrased to simply convey cervical factors rather than a specific characteristic of the cervical exam.

Response: We agree and changed accordingly. "These findings may suggest that uterine contractions with lower MAR at the later stage of active labor are sufficient to remodel the cervix effectively and rapidly in the multiparous subjects." (See Line 313-315).

**Key point.** What is a weak contraction by EMMI?

Response: We agree with the reviewer and removed the word “weak”, which is qualitative, and replaced it with lower MAR.

Discussion

TOCO was defined in line 39 and used throughout the manuscript, no need to repeat

Response: We agree with the reviewer and removed the definition.

an array of SQUID devices

Response: We agree with the reviewer and added “an array of ...”. (See Line 330).

perhaps would be clearer to the uninitiated if you used the phrase “EMG (also called electrohysterography, EHG)”

Response: We agree with the reviewer and changed it accordingly. (See Line 337).

If you wish to point out the limitation of SARA, you should also point out the advantage, specifically that MMG signals do not attenuate because of interfaces in the volume conductor.

Response: We agree with the reviewer and added: “...fixed pattern to collect signals from the anterior abdominal region without much attenuation and distortion from the interfaces in the volume conductor.” (See Line 332-333).

“all” is perhaps a bit hyperbolic. “Many at different locations” or similar, may be more accurate.

Response: We agree with the reviewer and removed “all of” as suggested.

**Key point.** “highly accurate” is an overstatement of the field. Uterine contraction tracings by EMG are an alternative to TOCO but have a much higher false positive reporting rate (compared with IUPC gold standard). This section of the discussion overstates the success of current EMG-based monitors. It would be more accurate to state that EMG can be used to produce a uterine activity tracing that emulates TOCO and is more reliable than TOCO with obesity, but discussion of the predictive capabilities of EMG should be entered into cautiously since this paper does not deal with any predictions.

Response: We agree with the reviewer and incorporated the suggestions into our revised discussion. The statement about the prediction was removed.

“EMG can be used to generate a uterine activity tracing that emulates TOCO and is more reliable than TOCO in patients with obesity.” (See Line 341).

reference 39 is erroneously referenced to TOCO (although it should be included elsewhere).

Response: We corrected the citation (Ref 42, [Young, R. C. Best Pract Res Clin Obstet Gynaecol (2018)]). (See Line 376).

**Key point.** If you say this, then you have to also say that the propagation “velocity” (really a speed) is controversial and then reference Mikkelsen et al 2013 Acta Obstet Gynecol Scand (plus perhaps a few others, including Catherine Marque, that refute these conclusions). Even though many authors state that they can diagnose true labor and false labor, it would be best to soften all these comments and state that some investigators have suggested that diagnoses can be made using EMG.

Response: We thank the reviewers for the information and suggestion. We agree with the suggestion and incorporate the following into our revised discussion.

“Although some EMG studies have revealed that the electrical propagation velocity increases at active labor compared to that at non-active labor²⁸, the previous studies also reported controversial findings. [Mikkelsen E. et al 2013 Acta Obstet Gynecol Scand, 2013(92)]”. (See Line 345-546).

334 – 5. **Key point.** Same, stating that any feature of EMG signals can identify the onset of labor is not true. This ability has never been shown in prospective clinical trials.

Response: We thank the reviewers for the information and suggestion. We agree with the suggestion and incorporate the following into our revised discussion. “Additionally, several features of EMG signals, such as the intensity, peak frequency of power spectrum, etc. show some promise in identifying the onset of labor.” (See Line 347-348).

it wouldn’t be impossible, just not ethical.

Response: We used “unethical “to replace “nearly impossible”. (See Line 356).

high spatial and temporal

Response: We used “high spatial and temporal”.

not sure why “time domain” is included here and not the prior sentence. What is a counterpart? Is it the source of surface EMG? These 2 sentences are confusing and should be re-written to focus.

Response: We re-written the sentence. “EMMI can also image the uterine surface electrograms to reflect the local uterine electrical activities (Fig. 3).” (See Line 362-363).

One of the strengths of EMMI is that it clearly delineates the myometrial regions that fail to activate during a contraction.

Response: We agree with the reviewer’s comment.

not active within the sensitivity of our recordings. At this point, a discussion as to why the back side of the uterus appears to activate less than the front. Is this a volume conductor effect? See below.

Response: We thank the reviewer for the comments. We also have the same observation in our subjects with good quality MRI and electrical recordings. Therefore, we believe the results are true based on the best performance of our current version of the EMMI system. However, we also agree that the distance between the lower back and posterior uterus is longer than the distance between the anterior abdomen and anterior uterus, which creates asymmetric geometry of the volume conductor delineated by the uterus and body surfaces and leads to the lower electrical signal SNR on the back. This could make the conventional fixed penalty term selection in the Tikhonov inverse computation less optimal, especially for the inverse computation of the potentials on the posterior uterine wall. A better inverse algorithm with an unfixed penalty term will help to further confirm our findings in future development.

– “and” should be “or” body surface EMG without EMMI or other methods of mapping involving inverse solutions.

Response: We changed “and” to “or”.

new paragraph – Distinct..

Response: We started a new paragraph from “Distinct...”.

361 -3 change “Interestingly” to Our results indicate that ... and is direct evidence against anatomically fixed ... add here [ref 39]

Response: We made the suggested changes. (See Line 373-376).

human uterine contraction [add reference 39, 44 and Young, Reproduction 2016 152(2), R51-61]

Response: We made the suggested changes and add the references. (See Line 377).

365 - 6 Not so sure about information obtained over the posterior of the uterus. Suggest you soften this sentence a bit.

Response: We made the changes to soften the conclusion: “This human EMMI study thus demonstrates that EMMI can noninvasively provide rich information on human uterine surface activation patterns.” (See Line 377-378).

“Imaging imaged”. One or the other. Actually, the strength of the isochrome map is that incorporates both the timing and extent of activation into a single image. The “entire” aspect has already been emphasized.

Response: We made the corresponding changes “EMMI isochrome maps provide detailed activation sequences over the activated uterine regions.” (See Line 379).

– omit “rich”

Response: We removed “rich”.

“times during” would sound better as “over the course of a contraction”.

Response: We thank the reviewer for the suggestion and made the suggested change. (See Line 381).

see comments above regarding SYN – as you calculate it , SYN is a rate and doesn’t reflect the extent.

Response: We replaced SYN with ACS and made corresponding changes in the revised manuscript.

**Key point:** cardiac synchronization is different since all parts of the heart contract with each heartbeat and there is organ-level synchronization by an electrical activation mechanism. You have clearly shown that not all parts of the uterus contract each time and that organ-level synchronization is not by electrical activation (probably is by mechanotransduction, see Young, 2016 Reproduction). There is no reason to define SYN based on cardiac literature. Now that I see why you defined SYN this way, I can now respectfully disagree that this definition adds information to uterine physiology. SYN possibly could be measured as either elapsed time to MAR or simply MAR itself (see comments on analysis of sigmoid curve, below).

Response: We thank the reviewer for the suggestion, and we agree with the reviewer and replaced SYN with the activation curve slope (ACS).

and the refractory regions are exactly why understanding the falling phase of the contraction and the time between contractions is important. There is no information on refractory times (or durations, rather) in this work, but since you clearly state “potentially explained”, this sentence can stand.

Response: We agree with and thank the reviewer’s comment.

**Key point:** See comments above on downward. This will require a lot of explaining if you wish to use the word “downward” in the discussion. As it stands, I cannot see that your data supports a downward movement or contraction pattern. The uterus is a pressure-generating organ, it does not use peristalsis or catastalsis to dilate the cervix. This is a common misconception, especially among clinicians and since your data argues against this, this discussion would be a good place to help stop this misconception.

Response: We agree with the reviewer. The uterine surface electrical pattern observed by EMMI is not sufficient to infer uterine mechanical movement. The electrical-mechanical-fluid coupling will need to be considered in the future to draw such a conclusion. We removed “downward” accordingly in the revised manuscript.

Where does your data support fundal dominance? Patient 4 is the only subject that shows the fundus activates early in all three contractions. Keep in mind that the fundus is thicker than the lower uterine segment and the number of myocytes per surface area is skewed towards the fundus. Thus, if each myocyte is a potential “pacemaker” for a particular contraction, then your 25% definition of fundal by surface area would likely be the site of origination 30% of the time based purely on probability.

Response: We agree with the reviewer and changed the sentence to “to reflect the uterine contraction pattern...” (See Line 398).

contraction during normal and arrested labor

Response: We made the suggested change. (See Line 398).

again, the MAR show greater synchronization effects better than SYN, but where does the fundal dominance/ downward statement come from?

Response: We removed “more downwards contractions”.

the duration of labor is associated with success of vaginal delivery (labor progression) but it is likely not causative. I think you are trying to get at the ability to separate dysfunctional uterine contractility and cervical factors with arrested labor. This discussion section on oxytocin doesn’t add a lot, since you didn’t give any clinical data on reason for induction, how much oxytocin subjects were on, how long they were on, and the cervix exam at the start of induction. These clinical variables are critical and there are

too many to evaluate with only 5 patients in each group (spontaneous versus oxytocin, not multip/nullip). Please reconsider discussing oxytocin in depth.

Response: We agree with the reviewer, and we do not have sufficient patient and clinical information to discuss the oxytocin effect in depth. We removed the following sentence in the discussion “For example, for patients with slow labor progression but strong uterine contraction (Fig. 4b), administration of oxytocin may not be sufficient to provide for labor progression.”

Myometrial memory is a clever clinical concept, but without any basis in physiology. It is more likely that the cervical compliance of multips is the difference based on restructuring after 10 cm dilation.

Response: We agree with the reviewer, in addition, reviewer #1 also suggested we cite one paper, Govindan et al Reprod Sci 2015 May;22(5):595-601. doi: 10.1177/1933719114556484. Epub 2014 Oct 27. to support the “myometrial memory” observation.

how do you define weak versus strong using EMMI? I haven’t seen that in your manuscript yet.

Response: We changed the weaker to “with lower MAR”. (See Line 417).

“Can be” You haven’t previously mentioned any of these complications, and none of your subjects had any of these conditions. If you wish to refer to abnormalities in general, that is fine, but perhaps should avoid proposing clinical trials for specific diseases without justification.

Response: We agree with the reviewer. Those are rough discussions in general and as the reviewer pointed out, we are not proposing clinical trials for any specific diseases. We changed “can” to “could”. (See Line 465).

Missing from discussion (each is a key point)

- **Key point:** Sigmoidal curve for activation over time. This addresses the same physiology as the SYN, which I believe is the slope at the inflection point of the curve. These types of curves have been classically addressed using the Hill equation, or similar, and that type of analysis would provide a more comprehensive physiological assessment of the data. The interpretation of the data would also be more relevant to the uterus, rather than forcing a cardiac parameter that relates to organ-level action potential propagation (which does not occur in the uterus)

Response: We agree with the reviewer and thank the reviewer for pointing us to the Hill equation to better describe the uterine activation curve derived by EMMI. We also add the discussion of the Hill Equation related to the sigmoidal activation curve derived by EMMI, and its potential to provide a more comprehensive physiological assessment of the EMMI data with more relevance to uterus contractility. See below:

“In this first human EMMI study, we derive the intuitive EMMI indices such as MAR, ACS, and FAR, which relate to organ-level electrical activity patterns and propagation. We found that the EMMI-derived

human uterine activation curves have the sigmoidal evolution nature with respect to time, which potentially reflects the bioelectrical stimulation-response dynamics during uterine contractions [Hill, Archibald Vivian. *Journal. physiol.* 40 (1910).] An earlier in vitro study using the rabbit uterine muscle studied the relationship between maximum contraction tension and duration/strength of electrical stimulation [Csapo et. al. *The Journal of physiology* 126.2 (1954): 384.]. They found Hill's force-velocity relationship between the kinetics of isotonic contractions as a function of load (tension). Our data clearly confirm a potential regenerative process in human uterine contractility using a positive feedback mechanism. These findings will enable future studies to derive comprehensive and uterus-relevant physiological assessment of human uterine contractions using EMMI.” (See Line 429-439).

- **Key point:** In almost all subjects, the back of the uterus, and somewhat the sides, do not appear to be as active as the front. Is this more likely true, or more likely an artifact of measurement? The volume conductor of the back – including muscle/fascia/spine/longer distances is very different than the anterior abdominal wall and may decrease signal amplitudes so that the EMG signals do not rise above threshold in many cases.

Response: We thank the reviewer for the comments. We also have the same observation in our subjects with good quality MRI and electrical recordings. Therefore, we believe the results are true based on the best performance of our current version of the EMMI system. However, we also agree that the distance between the lower back and posterior uterus is longer than the distance between the anterior abdomen and anterior uterus, which creates asymmetric geometry of the volume conductor delineated by the uterus and body surfaces and leads to the lower electrical signal SNR on the back. This could make the conventional fixed penalty term selection in the Tikhonov inverse computation less optimal, especially for the inverse computation of the potentials on the posterior uterine wall. A better inverse algorithm with an unfixed penalty term will help to further confirm our findings in future development.

- **Key point:** Discussion of MAR as a function of cervical dilation would be good

Response: We thank the reviewer for the comments, and we add the following discussion to the revised manuscript.

“The clinical utility of EMMI-derived outputs will require prospective clinical trials involving multiple EMMI assessments throughout labor, for example, the correlation of MAR with cervical dilatation. In this initial study with 10 uncomplicated singleton pregnancies studied with EMMI in labor at 37w0d – 40w6d gestation and at cervical dilatation between 3.5 and 9.5cm, we observed that nulliparous subjects demonstrated different phenotypes from the multiparous subjects (Fig. 4 and 5). These initial results may be suggestive of subgroups of uncomplicated nulliparous women in labor.” (See Line 421-428).

- **Key point:** Uterine activities appear similar to surface EMG – Was the inverse solution necessary using these multiple pin electrodes? Would this be the same for larger BMI subjects.

Response: We agree with the reviewer that the body surface potential maps have correspondence and some similarity with the uterine surface potential maps because they are associated with one another

through a linear equation reflecting the body-uterus geometry. The inverse solution is necessary for EMMI for the following reasons: (1) There is a large inter-patient variance in the body-uterus geometry due to very different body sizes, shapes, and orientations of the uterus across different patients. Without the EMMI inverse solution, the physicians will have to conduct a “mental inverse computation” when they try to make a uterine region-specific interpretation. To do this, the physicians have to mentally integrate the body surface signals and patient-specific abdomen-uterus geometry, which is very difficult and prone to errors. As the reviewer pointed out, “one of the strengths of EMMI is that it clearly delineates the myometrial regions that fail to activate during a contraction.”. That is to say, EMMI will perform such an inverse solution for the physicians in an accurate and quantitative manner to provide noninvasive electrical information of the specific myometrial region for each subject. (2) The uterine surface EMG has better specificity because it is much closer to the true electrical source: the action potential in the myometrium cells.

In addition, a larger BMI can potentially reduce EMG SNR, which can potentially reduce EMMI accuracy. Similarly, the skin property is another confounding factor that can potentially affect EMMI quality. Better electrodes with larger surface areas and proper preparation of the skin before recording may improve EMMI quality. Toward this direction, we are developing various types of printed electrodes [Lo, L., et al., ACS Appl Mater Interfaces. 2021 May 12;13(18): 21693-21702. Lo, L., et al., ACS Appl Mater Interfaces. 2022 Feb. 14;14(7): 9570–9578.]

• **Key point:** In your early activation maps, is there an “earliest (or first) activation” site you can identify for each contraction. This would be a great help in definitively stating that there is no fixed pacemaker in the uterus.

Response: We thank the reviewer for the suggestion, and we labeled the earliest activation region using the yellow cross in Fig. 4 and Fig. 5.

Missing from the discussion but not a Key Point

• This group has also presented data on sheep. They are uniquely positioned to describe the similarities and differences of the physiology of labor in these two species.

Response: We thank the reviewer for this comment and suggestion. We also think this comparison will be of interest and plan to present it in a future manuscript.

Methods

umbilicus

Response: We revised based on the reviewer’s suggestion. (See Line 474).

“near” needs definition. If this is placed by an OB clinician as located at the fundus, state so. If placed by a non-obstetrician, state how located and how trained.

Response: We revised based on the reviewer's suggestion. We added the following sentence in the revised manuscript. "The fundus was located by an obstetric clinician". (See Line 475-476).

coccyx. So where were the tapes placed, using these landmarks?

Response: We revised based on the reviewer's suggestion.

"Similarly for the back surface, a measuring tape of 30 cm was placed vertically along the spine and ending at the coccyx; another tape of 61.3 cm was placed horizontally at the upper edge of the ilium over the vertical tape." (See Line 476-478).

"...(Fig. 1). For the abdominal surface, a measuring tape of 30 cm...". Mention the lengths of the tapes (30 cm and 61.3 cm). (See Line 473-477).

phrasing typographical error

Response: We moved "192" before MRI-compatible markers. (See Line 479).

We assume that the electrodes refer to the EMG sensors/electrodes used later?

Response: Yes, and we added "mimic the electrodes for electrical recordings during active labor". (See Line 480).

"or" or "and" – was only one image obtained (this is how the sentence currently reads, but is confusing). If one image, use of "either" would help.

Response: We revised based on the reviewer's suggestion. (See Line 481).

Fowler was a person and should be capitalized (Butch would have been happy to lend his name here, but this description is most commonly used in gyn oncology and is not common obstetrical parlance, or to the readers of this journal. Probably semi-recumbent with legs elevated would be better)

Response: We capitalized "Fowler". (See Line 493).

While most readers know what an AD-box is, better to write out analog to digital converter and whether it is custom or commercial. I assume the details are given in the sheep publications, so a specific reference would also clear this up.

Response: We replaced "AD-box" with "analog to digital converter". (See Line 494).

Key point: Needs to include bandpass for recording, where ground was placed if bipolar, or if monopolar recordings. If not included later, how signal filtering was accomplished, Butterworth, etc., and precise frequencies.

Response: We revised based on the reviewer's suggestion. Specifically, we added "Four ground electrodes were placed at left/right upper chest and left/right lower abdomen, respectively." (See Line 488-489).

Signal filtering was described in the section "Abdominal surface EMG signal preprocessing".

Software

simplify to "processed using Matlab software". The extra words you use are distracting and subject to misinterpretation. What does "custom" refer to – the Matlab software or the software you wrote using Matlab without modifying the Matlab programming.

Response: We revised based on the reviewer's suggestion. (See Line 498).

– "used to determine" or "delineate" the triangular meshes used in inverse calculations. Define is a squishy word.

Response: We replaced "defines" with "delineates" based on the reviewer's suggestion.

OK, fine to put Butterworth here

Response: We than the reviewer for the comment.

**Key point.** Yikes, why 0.34 to 40? I see you drop to 1 Hz later, but the 40 Hz will capture skeletal muscle. If these frequencies were used in your noise reduction algorithm, please state, otherwise this section is simply confusing. Also, the 0.34 – was this to reduce respiration? Most in the field use 0.2, and some as low as 0.1. The uterine bandpass is so narrow to begin with, narrowing to 0.34 to 1.0 bandpass may loose information. Please justify. However, if you feel this bandpass successfully identifies on/off uterine activity, which is the goal of the paper, then state that clearly and note that others have used slightly wider.

Response: We revised based on the reviewer's suggestion.

40Hz low pass filter aims to prepare the EMG signal for the 100Hz down-sampling since the original 2kHz sampling rate is redundant for the uterine EMG analysis. The second sentence described the analysis frequency range is 0.34 - 1Hz. The 0.34 Hz is to reduce the respiration artifacts, which is less than 0.34 Hz. The 0.34 - 1Hz is also widely used in the field and has been reported to reveal the main feature of uterine contractions [Vasak B, et. al. Am J Obstet Gynecol. 2013 Sep;209(3):232.e1-8].

We added the statement as follows: “The 0.34 Hz high-pass filter aims to reduce the respiration artifacts [Vasak B, et. al. Am J Obstet Gynecol. 2013 Sep;209(3):232.e1-8], which otherwise will affect the accuracy of identifying the onset of the EMG burst. ” (See Line 510-511).

Why down sample here – an anti-alias attempt?

Response: The original sampling rate in the raw electrical recording from the BioSemi system is 2kHz, which is high and redundant for uterine EMG analysis. The 100Hz down-sampling will significantly reduce the data size and computation time of EMMI.

to 477 Rephrase this section. It probably says what you mean to say, but takes three readings to really understand.

Response: We revised based on the reviewer’s suggestion.

We revised the sentence as follows: “Finally, a multi-step artifact detection algorithm was applied to the band-passed signal to detect invalid EMGs containing abnormally large EMG data and distorted abdominal surface potential maps (ASPM). The first quartile of the mean absolute magnitudes of each processed EMG was a reference, any EMG with an absolute magnitude larger than 100 times the reference is detected as an invalid EMG. The median values of the mean absolute magnitude of all ASPMs were a reference, and any ASPMs with mean absolute magnitudes larger than ten times the reference are detected as distorted ASPMs. (See Line 513-525).

“moving” refers to window and is misplaced

Response: We revised based on the reviewer’s suggestion. (See Line 520).

How about EMG signal obscured by artifact, rather than bad. Was it exactly half, if so say 50%.

Response: We revised based on the reviewer’s suggestion.

If the EMG signal obscured by artifacts satisfies the aforementioned criteria, it will be defined as an invalid EMG. Half refers to 50%. We revised the description as follows: “A BSPM with more than 50% of the signal ... An ASPM with more than 50% of the sites contaminated by ...” (See Line 523-524).

using Thermo

Response: We revised based on the reviewer’s suggestion. (See Line 529).

using Artec... to precisely determine electrode locations

Response: We revised based on the reviewer's suggestion. (See Line 529).

to 8 stilted phrasing – reword

Response: We revised based on the reviewer's suggestion. (See Line 528-529).

change derived to obtained from the MRI and 3D optical scans.

Response: We revised based on the reviewer's suggestion. (See Line 528).

Normal and equal length projections were calculated using a customized algorithm written using Matlab.

Response: We revised based on the reviewer's suggestion.

We reworded it as follows: "Second, align the 3D optical body surface to the MRI-derived surface and register the 3D optical electrode locations onto the MRI-derived surface (see Supplementary Fig.1 for details)". (See Line 529-531).

not sure what landmarks we are talking about here, clarify.

Response: We revised based on the reviewer's suggestion.

The landmark here represents a specific geometry mark in Amira software to acquire the coordinate of the electrode. We revised the sentence as follows: "3) Obtain the coordinates of electrodes and the points on uterine mesh surface to construct the abdomen-uterine geometry." We reorganized the section based on the comment in line 500 (see below).

For the uninitiated reader, please imply why all this was performed, such as to completely define the spatial relationship between the uterus and skin (the medical definition of "abdomen" is not how you use it here)

Response: We implied the why in the first sentence and reworded the whole paragraph after. We revised based on the reviewer's suggestion. We used "body-uterus" in the revised manuscript.

We revised the description as body-uterus geometry. This section/paragraph is to introduce the method used to generate the body-uterus geometry, as stated in the title. To make it clear, we reorganized the section as follows: "The sagittal slices of MR images and 3D optical scans are used to generate the body-uterus geometry. There are three main steps: 1) Obtain the triangulated mesh surfaces from MR images and the 3D optical scans, separately using Amira Software 6.4 and Artec Studio 12. 2) Align the 3D optical body surface to the MRI-derived surface and register the 3D optical electrode locations onto the MRI-derived surface (see Supplementary Fig.1 for details). 3) Obtain the coordinates of electrodes and the points on the uterine mesh surface to construct the body-uterine geometry. " (See Line 527-533).

Inverse computation

to 521 This reviewer is unable to adequately evaluate this section because it is outside his level of expertise. That said, it is important to clearly state if any human intervention was required to optimize or identify the optimal inverse solution, and if the inverse solution always led to a unique result and the algorithm was well-behaved. Clearly state that no human intervention was required, if that was the case.

Response: We thank the reviewer for the comment. We rephrased the statement as follows: “The Tikhonov regularization was employed to stabilize the ill-posed inverse computation, which gave a unique solution for each measured abdominal surface potential (ϕ_U), and no human intervention was required.” (See Line 550-552).

Data visualization

to 525 **key point** – need precise time resolution stated here rather than “instant”. An approximate value is acceptable here but needs to be defined somewhere.

Response: We changed “instant” to “time point” and mentioned time resolution right after as follows: “First, a uterine surface potential map is the electrical potential distribution on the 3D uterine surface at each time point. The time resolution is 102.4 Hz. Second, ...”. (See Line 555-557).

**Key point.** The phrase “uterine activation” is a bit of a problem. It appears that the authors mean sequential bioelectrical activity. However, the literature in the field has traditionally used the term “myometrial activation” to mean the conversion of the tissue from quiescent to spontaneously active – a process that takes days or weeks. Simply to avoid this mixing of terminology (which may be long-lasting), perhaps the authors should consider substituting a phrase like, “uterine electrical activation”, or “uterine bioelectrical activation”.

Response: We changed “uterine activation” to “uterine electrical activation” throughout the manuscript.

“EB”, or “electrogram burst” was defined on line 92 and while used extensively throughout the manuscript, appears abruptly in the methods section. Purely for readability, it would be best to assign (back at line 92) an abbreviation more closely associates with occurring on the uterus and delineate it from an electrical burst that is observed at the skin. This could be uterine surface signal, source signal, or a variety of other more specific phrases. Seeing EB is mentally translated to electrical burst rather than uterine surface electrogram burst and requires effort to determine the true meaning of the abbreviation each time it appears. While this is a minor point, the definitions described in this manuscript may survive permanently, and care should be taken to assign highly descriptive acronyms when possible.

Response: We changed “electrogram burst (EB)” to “uterine electrical burst (UEB)” throughout the manuscript. (See Line 98).

529. Again the activation problem, probable better to say uterus became electrically active and returned to inactive at that location.

Response: We revised it to “electrically activated” based on the reviewer’s suggestion.

activation time (duration?) or time activation began.

Response: We defined “activation time” as “The term “uterine electrical activation time” or just “activation time” is used here to refer to the initiation time of the UEB”. (See Line 162).

TKEO needs a reference, the Teager energy operator is widely known, but not everyone knows the TKEO or the advantage of this over TEO.

Response: We added a reference [Solnik, S., et al., Teager-Kaiser energy operator signal conditioning improves EMG onset detection. *Eur J Appl Physiol* 110, 489–498 (2010)] on the reviewer’s suggestion. (See Line 566).

It appears that the 7 second window for RMS sets the time resolution for the sequential images. If this is so, please state here. The 1.01 value to define threshold implies a 1% rise – is this really the case that the noise level was so low.

Response: We revised it to “a moving window”. (See Line 567-568).

The 7-second window does not set the time resolution for the sequential images here but serves as a moving window (parameter) when deriving the RMS envelope from the absolute TKEO signal. The detection of uterine electrical burst (UEB) starts from the RMS envelope of the absolute TKEO signal, where signals between activation and baseline can be better distinguished. The 1% rise is on the top median of the RMS envelope, which means that more than half of the EMG signals are below the baseline level. This is a soft, data-driven, threshold, which means for good cases, the value is small and for invalid cases the value is large. Moreover, the noise threshold is defined as $\text{baseline} + 2 \times \text{std}$ of the baseline.

the baseline statement is either trivial or it is unclear what is meant.

Reading on, 536 to 539 taken as a whole is inconsistent and unclear.

Response: We reworded this part as follows: “... (the black line in Supplementary Fig. 2c), to distinguish signals of activation or baseline. The threshold for baseline was defined as 1.01 times the median of the RMS envelope signal (the blue line in Supplementary Fig. 2c). The threshold for electrical activation was defined as the mean plus twice the standard deviation of the baseline signals.”. (See Line 568-572).

Downsampling may affect temporal resolution and should be noted, also confusion over activation time phrasing comes into play again.

Response: We stated the time resolution and reworded the sentence to avoid confusion. The time resolution of the RMS envelope signals is 5Hz which is enough because EMG is 0.34-1 Hz.

regarded or defined as

Response: We replaced “identified” with “defined” based on the reviewer’s suggestion.

the EMG is the signal, and I think you are referring to a contraction here. (EMG identified contraction, perhaps)

Response: We revised based on the reviewer’s suggestion. We reworded it as “The EMGs uterine electrograms with SNR higher than 5 dB are regarded as qualified contraction signals”. (See Line 577-579).

This is the section that may be considered as requiring human input – or is this purely the result of the algorithm? After reading through this several times, it is possible to figure out what is done. A set up as to why you are doing this could be at the beginning of the paragraph, then point out what problem points do to the images, then the solution you came up with. This will make it much easier to read through.

Response: We added “using Matlab software” to indicate that it is an algorithm without human input and explained why doing this at the beginning of the paragraph to make it easier to read.

We started the paragraph with “the initiation times of electrical activation were generated from individual uterine surface electrograms at uterine sites, without considering the spatial connectivity and thus may not preserve spatial continuity in uterine electrical activation pattern. So, with the raw initiation times of electrical defined on the 3D uterine surface, a series of...”. (See Line 581).

ever been activated during that contraction – or recently , or some time. Refer to the figures.

Response: We added “during that contraction” based on the reviewer’s suggestion. (See Line 596).

SYN looks like the rate of synchronization by your definition - the slope. Please reconsider, and make sure you are saying what you mean to say. For the uterus, synchronization is usually what fraction of the potential sources are active at the same time. Additional comments on SYN are in the discussion.

Response: We made changes based on the reviewer’s suggestion above and used ACS to replace SYN in the revised manuscript.

Extended data figure 1

In general, this figure legend isn't clear. The purpose of the registration is to define the spatial relationships between the uterine surface and the skin. Ideally this would be able to be visualized using only the images, but it is very difficult to do this. While this figure may be technically correct, it could be improved to explain the process more simply.

Here are some specific places

"a" looks like an overlay of green and blue, colors aren't clear. "showing MRI-safe marker locations" (MRI-safe is redundant information for this audience)

Response: We followed the editorial requirement to replace "extended data" with "supplementary".

We deleted "MRI-safe" based on the reviewer's suggestion. We remade Supplementary Fig. 1. (See Line 771)

Does green represent the results of the procedures that align (or correlate) uterine surface locations with abdominal surface locations?

Response: We remade the figure. In Supplementary Fig. 1b, grey represents the aligned optical-scanned abdominal surface with electrodes (blue) on it.

it isn't clear the difference between rigid and non-rigid.

Response: We replaced "non-rigid alignment" with "electrode registration" to avoid misunderstanding. Supplementary Fig. 1d shows the registration methods of electrode locations.

While the author's choice is technically correct, "orthogonal projection", or perhaps "projected to the nearest location on the skin" would have been more clear than normal, which is commonly used to mean "usual".

Response: In geometry theory, normal direction means the direction that is perpendicular to the tangent plane on the curved surface. "Orthogonal projection" is a means of representing three-dimensional objects in two dimensions, while "projected to the nearest location on the skin" looks for the closest neighbor without actual directional projection in geometry. Therefore, we believe both terms don't describe our method accurately.

Extended data figure 2. This shows processing to the uterine activity (C) of a single channel and perhaps having a summation of channels as (D) would make it look like a TOCO emulation and be more clear. Not sure it is best to use mV on the ordinate for B or C since the energy operator has derivatives in it and not sure that mV adequately represents this.

Response: We changed “mV” to “Energy” on the ordinate for panels b and c based on the reviewer’s suggestion. (See Supplementary Fig. 2b and 2c).

It looks like getting to the blue boxes are the goal of this figure, but it would help to put an arrow or something that relates back to observing the activation you use in the text.

Response: We added green arrows pointing at the initiation of activations in the figure based on the reviewer’s suggestion. (See Supplementary Fig. 2c).

Key point: The green line does not look like 1% above the baseline as described in the text (1.01, line 536). It looks more like you defined baseline as the maximum value (1.01 times max?) observed between bursts.

Response: The blue line (green in the old version) is defined as the 1.01 times median of the RMS envelope of the absolute TKEO. In the 170 seconds of the RMS envelope (see panel c), about 125 seconds were low baseline signals while about 45 seconds were high activation signals. Based on the definition of median value from the following theoretical example: the median of (0,1,2,3,4,998,999) is equal to the median of (2,3,4) and is equal to 3. Similarly, the RMS envelope’s median is equal to the median value of around 80 seconds (=125-45) of baseline signals with the highest magnitude because the lowest 45 seconds baseline and the 45 seconds activation signals will not impact the median value. Therefore, 1.01 times of the RMS envelope’s median value will be above the median of the baseline. It just happens to look close to the maximum value of the baseline.

Reviewers' Comments:

Reviewer #1:

None

Reviewer #2:

Remarks to the Author:

All questions successfully addressed. Two technical changes suggested:

1. reference 42 is duplicated reference 40, somehow your endnote didn't catch.
2. The authors answered this reviewer's question regarding average size of regions (they state 8 to 14 cm, with plans to investigate further). If this range is based on an analysis - as opposed to just a guestimate - putting this into the manuscript would help support the Arkansas SARA results, since the data seem to agree despite using different recording technologies and different inverse solutions. If the authors agree, line 410 would be a good place to add a sentence noting this approximation/ estimation and state general agreement with SARA.

That said, these minor revisions are at the discretion of authors. Roger Young

Itemized Author Response Letter (Reviewer#2)

Reviewer #2 (Remarks to the Author):

1. reference 42 is duplicated reference 40, somehow your endnote didn't catch.

Response: We thank the reviewer for the comment. We fixed the duplication.

2. The authors answered this reviewer's question regarding average size of regions (they state 8 to 14 cm, with plans to investigate further). If this range is based on an analysis - as opposed to just a guesstimate - putting this into the manuscript would help support the Arkansas SARA results, since the data seem to agree despite using different recording technologies and different inverse solutions. If the authors agree, line 410 would be a good place to add a sentence noting this approximation/ estimation and state general agreement with SARA.

Response: We thank the reviewer for the comment. The range of 8 to 14 cm is just an estimation based on the limited number of cases, rather than a solid analysis. So, we decided not to make a general conclusion in this pilot work. We may further investigate the propagation range in future work.